

# Development of a Multichannel Organics *In situ* enviRonmental Analyzer (MOIRA) for mobile measurements of volatile organic compounds

Audrey J. Dang[1], Nathan M. Kreisberg[2], Tyler L. Cargill[1], Jhao-Hong Chen[1], Sydney Hornitschek[1],
Remy Hutheesing[1], Jay R. Turner[1], Brent J. Williams[1]

[1]Energy, Environmental, and Chemical Engineering, Washington University in St. Louis, St. Louis, 63130, United States
[2]Aerosol Dynamics, Inc., Berkeley, 94710, United States.

*Correspondence to*: Brent J. Williams (brentw@wustl.edu)

**Abstract**

Volatile organic compounds (VOCs) have diverse functionality, emission sources, and environmental fates. Speciated measurements of their spatiotemporal variability are thus key to understanding their impacts on air quality, health, and climate. Networks of passive samplers can be used to map VOC concentrations, or *in situ* instruments can be deployed on mobile platforms. Limitations of existing *in situ* instruments include high cost, identification of non-target species, differentiation of isomeric species, or low time resolution, which limits how quickly an area can be spatially mapped with mobile measurements. This work describes the development of the Multichannel Organics *In situ* enviRonmental Analyzer (MOIRA), which has been designed for *in situ* mobile measurements of target and non-target VOCs from the cargo area of a hybrid hatchback wagon vehicle. Staggered sample collection and analysis by four thermal desorption collectors, four miniature gas chromatography (GC) heaters, and two compact residual gas analyser (RGA) mass spectrometer (MS) detectors enable continuous measurements at 10 min time resolution. Non-target species and structural isomers can be identified with electron ionization (EI), and species detected include alkanes (from pentane to pentadecane) and aromatics, as well as more oxidized species such as aldehydes, esters, and carboxylic acids. The instrument is characterized in the laboratory under different environmental conditions and in two pilot field studies of indoor air in a single-family residence and of ambient air during a mobile deployment.

## 1 Introduction

Emitted by diverse sources including vegetation, combustion, traffic, cooking, cleaning, and volatile chemical product production and use, volatile organic compounds (VOCs) are subject to chemical reaction and dilution, resulting in large spatial and temporal variability (Andreae, 2019; Arata et al., 2021; Coggon et al., 2021; Healy et al., 2022; McDonald et al., 2018; Sindelarova et al., 2014; Stockwell et al., 2021; Yacovitch et al., 2015). As precursors for ozone and particulate matter, VOCs affect air quality, climate, and human health (Burnett et al., 2018; Coggon et al., 2021; Donahue et al., 2012;



Kampa and Castanas, 2008; US Environmental Protection Agency, 2019). In addition, VOCs are among the list of hazardous air pollutants (HAPs) designated by the US Environmental Protection Agency (EPA) (Initial List of Hazardous Air Pollutants with Modifications). Characterized by high volatility and their presence in the gas-phase, VOCs encompass diverse chemical functionalities and structures, and their influences on human health and atmospheric chemistry are highly dependent on precise molecular identity (Glasius and Goldstein, 2016). For a large fraction of VOCs that have been

measured in indoor and outdoor environments, investigations of health effects are limited to *in silico* studies in part due to challenges in estimating human exposure, which is affected by the spatial and temporal variability of individual VOCs in both indoor and outdoor environments (Hodshire et al., 2022).

To characterize spatial variability, passive VOC samplers with sorbents can be deployed throughout an area as a network prior to offline analysis (Amini et al., 2017a, b, c; Lu et al., 2019). The resulting spatially resolved measurements

can be used to create models for ambient exposure. To collect sufficient sample quantities, passive samplers are often deployed over days to weeks, though this time can be decreased to minutes to hours if actively pulling air through the sampler with a pump. Canister samples can also be collected at different locations and analysed off-line. These sampling methods have also been used to measure indoor air, but the lower time resolution is not well suited to characterizing dynamic changes, especially from activities such as cooking and cleaning (Chin et al., 2014; Farmer, 2019; Fortenberry et al., 2019;

Sax et al., 2004; Su et al., 2013).

Alternatively, *in situ* instruments with higher time resolution can be deployed to indoor environments and on mobile platforms (Arata et al., 2021; Liu et al., 2019). For example, the proton transfer reaction – mass spectrometer (PTR-MS) has been deployed on mobile research labs for spatial mapping, hotspot identification, fenceline monitoring, and plume chasing (Gkatzelis et al., 2021; Healy et al., 2022; Herndon et al., 2005; Knighton et al., 2012; Kolb et al., 2004; Warneke et al.,

2014; Yacovitch et al., 2015). While its time resolution (on the order of seconds) and high sensitivity (especially with a time-of-flight (TOF) detector) are well-suited to ambient mobile measurements, the PTR-MS is costly and operationally complex. Alternatively, a selected ion flow tube mass spectrometer (SIFT-MS) can only measure a more limited set of target analytes, but is simpler to operate (Wagner et al., 2021). However, both of these soft ionization techniques can be challenged by interferences between structural isomers, though this can be mitigated by a thermal desorption (TD) gas

chromatography (GC) inlet, which separates different compounds but greatly reduces time resolution (Claflin et al., 2021).

TD-GC techniques enhance sensitivity with preconcentration prior to thermal desorption and sample transfer to a GC column, but this lengthy process results in lower time resolution. For example, Photochemical Activity Monitoring Sites (PAMS) use *in situ* TD-GC with an FID detector for hourly VOC measurements (US Environmental Protection Agency, 2019). However, FID detectors are vulnerable to coelution and cannot be used to identify non-target analytes. In contrast,

MS detectors with electron ionization offer more selectivity with characteristic ion fragmentation patterns. However, *in situ* TD-GC-MS instruments often use large, power-intensive GC and MS units or have large footprints that would require more specialized research vehicles to accomplish mobile sampling (Wernis et al., 2021). Miniaturization of MS and GC components has enabled the development of more portable systems. The person-portable Griffin 510 has an inlet for *in situ*



collection of air samples, but reported limits of detection vary, and issues with transfer of less volatile compounds such as
naphthalene have been reported to limit the analyte range (Rodriguez and Almirall, 2021; Torres and Almirall, 2022;
Fiorentin et al., 2020). In addition, the time required for GC separation by both the larger and more portable *in situ* GC-MS
instruments would limit the speed of mobile mapping of an area, unless multiples of the same instrument were deployed.

This work describes a new instrument, the Multichannel Organics *In situ* enviRonmental Analyzer (MOIRA), which
was developed specifically for mobile measurements and integrates multiple TD-GC-MS measurement channels in order to
continuously sample VOCs with a time resolution of 10 min. Four miniaturized GCs and two compact residual gas analysers
(RGA) mass spectrometers are integrated in a single instrument, which fits in the back seat and cargo area of a hybrid
hatchback Prius V and can be powered by the vehicle's high voltage battery via an inverter and transformer. MOIRA
measures VOCs in the pentane to pentadecane volatility range, is suitable for target and non-target measurements, and can
achieve detection limits in the 10 to 100 ppt range. In this work, the instrument is characterized under different
environmental conditions in the laboratory and with field measurements from a stationary deployment measuring indoor air
in a single-family residence and from a mobile deployment in a vehicle in an urban environment.

## 2 Instrument and methods

### 2.1 Overview of MOIRA instrument

MOIRA integrates four sorbent traps, four miniature gas chromatographs, and two residual gas analyser mass spectrometers
into a single instrument with shared data acquisition, control, and vacuum systems (Fig. 1, Fig. 2). Sampling is continuous,
switching between the four sorbent traps (in order: A1, B1, A2, B2) every 10 minutes (Fig. 3). Each pair of sorbent collector
and GC channels (A1 and A2, B1 and B2) share one of two quadrupole residual gas analysers (RGA A, RGA B; Pfeiffer
Vacuum PrismaPro QMG250 with cross-beam source) for detection with electron ionization (70 eV) mass spectrometry.
Ten-port rotary valves (EUH-3C10WE, Valco) switch each pair of channels between sample loading (i.e., collection,
purging) and sample injection (i.e., thermal desorption, GC analysis), each of which last 20 minutes.

The RGAs are controlled with PV MassSpec software (Pfeiffer Vacuum), which acquires the MS data. Sample
collection and gas chromatography are controlled by a custom LabVIEW (National Instruments) program run on a small
form factor computer (Intel NUC 8I5BEH).

The physical dimensions of MOIRA are 1.22 m length, 0.58 m width, and 0.65 m height (including the instrument
chassis and its wheels, but not the helium gas cylinder). A single ultra-high purity helium cylinder (size 300) is used for
purge gas, calibration matrix in the field, and carrier gas. A rigid scaffold including the vacuum chamber, MS detectors,
GCs, sample collectors, and computer is mounted to the instrument chassis with four helical vibration isolators (Enidine
WR6-950-10).



(a)

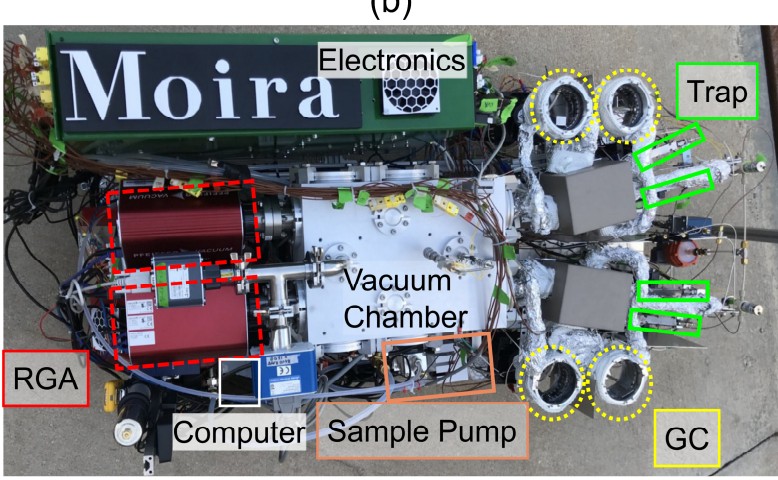

(b)

Figure 1. (a) The MOIRA instrument during a deployment from the cargo area of a Prius wagon vehicle. (b) Overhead view of the instrument.

95



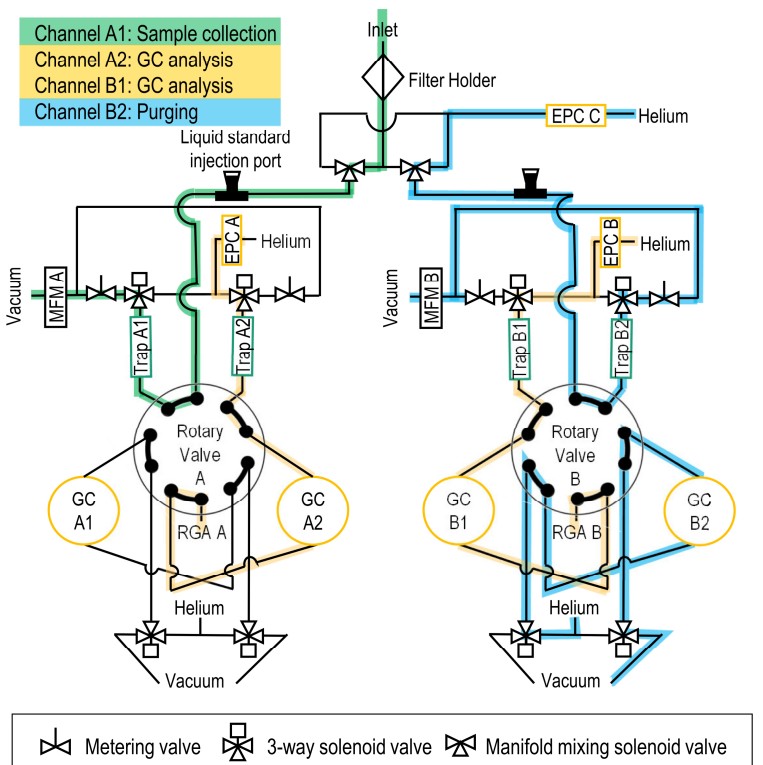

**Figure 2. Flow diagram for MOIRA instrument with flow paths indicated for sample collection by Trap A1, GC analysis by GC A2 and GC B1, and purging of Trap B2 and GC B2. EPC = electronic pressure controller, GC = gas chromatography heater with 30 m metal capillary column, MFM = mass flow meter, RGA = residual gas analyser mass spectrometer, Trap = multibed sorbent collector.**

100



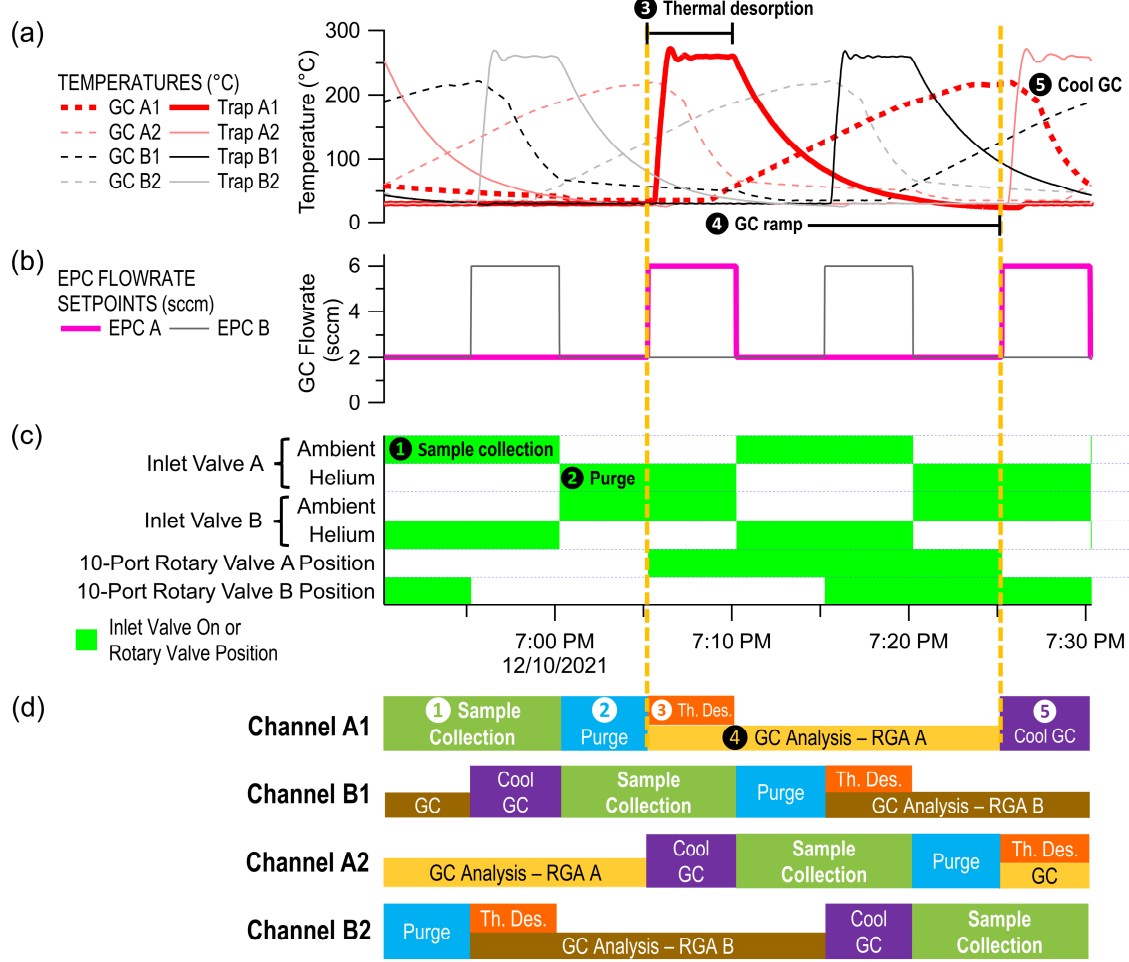

**Figure 3.** Schematic of the sample collection and GC method with numbered steps for channel A1. The methods of the four channels are staggered in time such that the 10 min sample collection periods do not overlap.

## 2.2 Sample Collection and Thermal Desorption

At any given moment during ambient measurements, one of the four sorbent collectors is sampling ambient air in non-overlapping, 10 min collections (70 mL min$^{-1}$). Previously described by Wernis et al. (2021), the sorbent collectors and their heating blocks (Aerosol Dynamics Inc.) feature low swept volume and low thermal mass, enabling direct, fast sample transfer from the collector to the GC without an intermediate focusing collector. To enable quantitative sample collection



and transfer, Inertium® coating (AMCX, Inc) is applied to all stainless steel surfaces in the sample flow path that are heated above ambient temperature.

The combination of sorbents in this same multibed collector has been previously shown by Wernis et al. (2021) to collect VOCs in the pentane to pentadecane range. In the direction of sample flow during collection, the order of sorbents is glass beads (50 mg, 40-60 mesh, Sigma Aldrich), Tenax TA (10 mg, 60-80 mesh, SIS Instruments), glass beads (10 mg, 250 µm diameter, Sigma Aldrich), Carbotrap B (20 mg, 60-80 mesh, Sigma Aldrich), glass beads (10 mg, 250 µm diameter, Sigma Aldrich), Carbotrap X (20 mg, 60-80 mesh, Sigma Aldrich), glass beads (10 mg, 250 µm diameter, Sigma Aldrich), and finally, glass wool (Sigma Aldrich) to fill the remaining volume of the collector.

The sample collection flowrates are set for each channel with metering valves (SS-2MG, Swagelok) based on measurements with a bubble flowmeter (Gillibrator-2, Sensidyne). The range of flowrate measurements across channels is typically 3 mL min$^{-1}$ at the most. A sample flowrate of 70 mL min$^{-1}$ was chosen as it was just below the maximum flowrate of the most restrictive of the four traps, some of which had maximum flowrates over 150 mL min$^{-1}$. The variability in the flow restrictions of the different traps is caused by differences in the packing, not in the collectors themselves, and increasing the uniformity of the glass wool packing could increase the similarity of the trap restrictions. Microbridge mass flow meters (AWM3100V, Honeywell) also monitor sample flowrate downstream of the sorbent collector.

Following sample collection, the sorbent collector is purged with ultra-high purity helium (28 kPa) for 5 min to remove water and oxygen. The hydrophobicity of the sorbents mitigates the collection of water. In addition, during collection, the sorbent collector temperature is maintained at 30 ºC, which is higher than ambient dew point for a wide range of field conditions.

After purging, the collected analytes are thermally desorbed and transferred to the GC column. The ten-port rotary valve position is switched, and the sorbent collector is backflushed with helium at 6 mL min$^{-1}$ while being rapidly heated from 30 ºC to 260 ºC, which is maintained for 4.5 min. Due to the low thermal mass of each trap and its heating block relative to the power of their two cartridge heaters (100 W each), 80% of this temperature increase is achieved in 35 to 40 s.

**2.3 Gas Chromatography**

The collected VOCs are separated by gas chromatography with a metal capillary column (MXT-624, 30 m length, 324 µm ID, 1.8 µm phase thickness, Restek). Each of the four GC columns is wrapped in a single layer around the exterior of a miniature cylindrical hub (Aerosol Dynamics Inc), which has a diameter of 8.5 cm and a height of 6.0 cm (Wernis et al., 2021). Heavy aluminium foil (McMaster 9012K26) and fiberglass insulation are layered on top of the wrapped columns. A 150 W mica band heater (2.5 cm length, 7.6 cm expanding diameter with wedge lock) contacts the interior surface of the hub with a water-soluble heat transfer cement (Tracit-600A, Chemex). The K-type thermocouples for the four GCs have a one-point offset calibration with an ice water bath.

The carrier gas flowrate is held at 6 mL min$^{-1}$ during thermal desorption and then lowered to 2 mL min$^{-1}$. After holding for the first three minutes of thermal desorption, the GC temperature is ramped from 35 °C to 172.5 °C in 10 min (13.8 °C



145   min$^{-1}$) and then to 215 °C in 4.5 min (9.4 °C min$^{-1}$). The GC temperature is held at 215 °C for 1.5 min. After the GC program is complete, the channel switches from sample injection to sample loading during which the GC column head pressure is a constant 138 kPa. The GC temperature setpoint is increased to 230 °C to purge less volatile compounds which might otherwise carryover between runs (back-flushing GC A1 and GC B2, and forward-flushing GC A2 and GC B1). The GC is then passively cooled until 5 min 15 s prior to the subsequent thermal desorption at which time the GC blower is powered on.

150   **2.4 Mass Spectrometry and Vacuum System**

The GC effluent is carried into the vacuum chamber by transfer lines (127 µm ID), which are parallel to the quadrupole rods and terminate just outside of the crossbeam ionization source of each RGA (Fig. S1). When compared to a transfer line orientation perpendicular to the hexagonal apertures and quadrupole rods, the parallel orientation had better sensitivity in limited testing, though the distance between the outlet of the transfer line and the source was not varied. The analytes are
155   ionized at 70 eV, and the ion fragments are detected in a scan from 20 to 200 amu at 2 ms dwell time per unit mass bin (total scan time of 553 ms, which also includes scanning and settling time). Alternatively, the RGAs can be operated in selected ion monitoring (SIM) mode at higher dwell times per mass bin to increase sensitivity to a smaller number of mass fragments which are abundant in the spectra of target compounds.

The RGAs are contained in a single vacuum chamber with two regions which are differentially pumped by two
160   turbopumps (Pfeiffer HiPace 300H) (Fig. S1). The source region contains the transfer line outlets and ionization sources, and the quadrupole region contains the quadrupole filters and electron multipliers. The conductance between the two regions is limited by custom ceramic sheaths and spacers which are mounted to each RGA, blocking the transmission of helium carrier gas from the source region to the electron multiplier region.

A plate separates the ionization sources of the two RGAs. For peaks of abundance above a threshold, infiltration of
165   analyte molecules through the gaps between the plate and the chamber walls can result in a "shadow peak" signal at the other RGA (Fig. S2). Comparing the relative magnitude of the two signals is simple and sufficient to identify this interference during data analysis. Compounds of higher volatility have higher magnitudes of interference (which ranges from 1 to 8%), suggesting that compounds of lower volatility (which would also have lower thermal velocity in vacuum if higher in molecular weight) are more likely to be adsorbed by surfaces, mitigating this effect (Fig. S3).
170   The vacuum system has hardwired interlocks that are independent of the computer and LabVIEW program, such that they can operate during start-up, shutdown, and maintenance as well as after a power outage. The vacuum gauge (MKS 390) must measure a chamber pressure less than 670 Pa for the turbopump motors (Pfeiffer HiPace 300H) to turn on. In addition, the RGAs and the hot ion gauge (MKS 350) do not receive power unless the chamber pressure is less than 0.13 Pa and the turbopumps have attained at least 90% of their setpoint rotational speed.





### 2.5 MS tuning and correction

A high vacuum solenoid valve (009-0270-900, Parker) and 100 µm orifice (SS-1/8-TUBE-100, Lenox Laser) control the introduction of perfluorotributylamine (PFTBA) into the vacuum chamber for tuning the quadrupole filters and electron multipliers of the RGAs.

Relative to NIST library spectra, ions with greater mass-to-charge ratios are less efficiently transmitted from the source to detector by the RGA's quadrupole (Stein et al., 2009; Ng et al., 2011). Sect. S3 describes how the mass spectra of multiple compounds in calibration standards are used to determine a relative quadrupole transmission efficiency correction for each RGA. The correction is not applied to the raw data, but rather can be applied to measured mass spectra prior to library search. While the cosine similarity and Pearson correlation of the MOIRA and library spectra are improved by the correction for several compounds, the match factor score used by the NIST library search is less impacted (Stein, 1994).

The electron multiplier voltages of the two RGAs are tuned for a gain of 1000 and further increasing this gain also increases signal noise, such that doing so would not increase sensitivity. When tuned, the overall signal of RGA A is at least three-fold higher than that of RGA B. Approaches for addressing this difference in sensitivity in data processing are described in Sect. 2.9, 2.10, and 3.5.

### 2.6 Power Requirements and Heater Control

MOIRA requires an average of 900 W of power during measurements and 600 W in standby. Maximum power consumption of 1100 W (as measured at 1 Hz) occurs during the first minute of thermal desorption (Kill A Watt EZ Model P4460.01). In deployments where minimizing power consumption is a priority, the sample collection and GC method can be modified to cool transfer line temperatures between samples and to use cooler transfer line temperatures, especially if the priority target analytes are more volatile.

MOIRA has 26 heated zones, which are controlled by the custom LabVIEW program using modified code from the Proportional-Integral-Derivative (PID) control toolkit (LabVIEW 2012, National Instruments). The heater duty cycles are staggered with a bin packing algorithm to minimize instantaneous power consumption. In addition, the Advanced PID VI has been modified to enable fractional decreases in the PID output for limiting maximum power consumption while limiting impacts to data quality and instrument operation. Finally, automatic heater shut-off upon detection of anomalies in a temperature zone's response detects operational irregularities and prevents catastrophic damage to components. These functionalities both give greater user control in field deployments during which power might be limited and support safe operation of the instrument, especially during mobile deployment. Further details are given in Sect. S1.2.

### 2.7 Liquid calibration standards

Liquid calibration standards were prepared in methanol (for gas chromatography, MS SupraSolv®, Supelco 100837) from pure components and purchased mixes. Given the potential for VOC calibration standards to evaporate over time during



storage, separate standards were prepared for the following: (1) indoor measurements and breakthrough tests (includes Accustandard M-8240A (Method 8240) and Accustandard VPH-WA-10X (aliphatic and aromatic mix)), (2) calibration curve, relative humidity, and ozone tests (includes Accustandard VPH-WA-10X (aliphatic and aromatic mix) and Accustandard CLP-022-PART-B) (VOC mix)), (3) tests of aldehydes and ketones with ozone (includes Accustandard M-554-R1 (aldehydes and ketones)), and (4) mobile measurements (includes Restek 30632 (VOC mix) and Supelco UST157 (alkanes)). All injections, including in-syringe dilutions, are 1 µL.

Liquid calibration standards can be injected via syringe at a 1/8" Swagelok tee with a Viton septum secured by a reducing port connector and held at either 65 °C (indoor measurements, breakthrough tests) or 75 °C (all other measurements). During laboratory calibrations, zero air is connected to the sampling inlet to evaporate the calibration analytes and carry them to the collectors. When zero air is unavailable during field measurements, ultra-high purity helium GC carrier gas is used instead, following the same flow path through the manifold mixing solenoid valves as the purge helium shown in Fig. 2.

Trap retention and breakthrough were tested by varying the time between calibration standard injection and thermal desorption (5, 7.5, 10, and 15 min). In addition, the effect of elevated ozone (100 ppb) and relative humidity (95% RH at 20 to 23 °C) were tested by alternating between the test condition (n = 3) and the field calibration method, which uses helium as the matrix gas (n = 4). The same test was done for dry zero air. The ratios of the peak areas for the test condition (elevated RH or ozone) and the average of the peak areas with helium matrix just prior and following were averaged for each condition. The effects of elevated ozone and RH with respect to dry zero air were calculated as the ratio of the averaged ratios for the environmental condition of interest (zero air with 100 ppb ozone or zero air at 95% RH) and the averaged ratios for the dry zero air condition.

A seven-point calibration curve was measured with a replicate of all concentration levels on each of six days. To correct for drift in instrument sensitivity, a correction factor for all compounds was determined from the peak areas from a single tracking concentration level T (the third largest of seven). First, each daily measurement of peak area $A_{T,ij}$ of each compound i on day j at the tracking concentration level T was normalized by the mean peak area $A_{T,i,mean}$ across all days for a given channel was calculated. Then, for each day j, the daily correction factor $C_j$ was calculated as the median ratio for all compounds at the tracking concentration level:

$$C_j = median\left(\frac{A_{T,ij}}{A_{T,i\,mean}}\right) \qquad (1)$$

All concentration levels for a given day could then be corrected for drift by normalizing by the daily correction factor. This correction factor better reflected instrument drift than measurements of the PFTBA tuning standard, likely because PFTBA abundance is sensitive to other factors including the flow of helium carrier gas into the ion source and potentially the temperature of the PFTBA vial and instrument.

An additional four-point calibration curve (n=1) was measured with a selected ion monitoring method to evaluate sensitivity at lower concentrations and higher dwell times (Table S1).




**2.8 Data Processing and Quantification**

Peak areas were quantified using the TAG ExploreR and iNtegration (TERN) package in Igor Pro (Wavemetrics) to integrate the peak areas of single ion chromatograms (Isaacman-Vanwertz et al., 2017). Compounds were identified by searching their mass spectra (with quadrupole transmission correction described in Sect. 2.5 and Sect. S3) with the NIST Mass Spectral Search Program for the NIST/EPA/NIH Mass Spectral Library (Demo Version 2.0f) and considering both search scores and retention time indices (Stein et al., 2009). Table S3 describes the confidence in compound identifications included in this work (i.e., confirmation with calibration standard, search score, etc).

Applying calibration curves is one approach to addressing differences in sensitivity between the channels, and both linear and power fits were investigated here. However, a compound may not be present in the calibration standards. Different data sets are amenable to alternate approaches as exemplified by the indoor and mobile measurements in the subsequent sections.

**2.9 Indoor Field Measurements**

From December 8, 2021 to December 11, 2021, indoor air was measured in a single family residence in St. Louis, Missouri. Various household activities were performed throughout the study including cleaning, cooking, door opening, and window opening (Table S4). The inlet of the sample tubing (10.8 m length, 6.35 mm outer diameter) was located in the kitchen with a 25 mm quartz filter at the instrument inlet.

Daily tracking standards were analysed on December 9, December 10, and December 11 for all four channels, and these responses were used to calculate a daily sensitivity factor that corrects for instrument drift over time as described in section 2.7. On December 11, an additional four concentration levels were analysed, and this multi-point calibration data was fit with a power law.

Since only a subset of compounds were in the calibration standard, compound-specific correction factors for RGA sensitivity differences were also calculated for each compound from the timeseries measurements themselves. First, RGA B peak measurements with non-zero RGA A peak measurements for that same compound in both the preceding and following chromatogram were identified. The mean of the adjacent RGA A peak areas was plotted versus the RGA B peak area. Measurements which had a Cook's distance of greater than four were removed, recalculating this parameter without the excluded data until all points met this threshold. A line through the origin was fitted for the remaining data, and the slope of this line was taken as the compound-specific correction factor for RGA sensitivity. If there were not sufficient measurements such that there were at least six measurements remaining after points in exceedance of a Cook's distance of four were removed, then the factor was calculated using a linear regression of the compound-specific factors for other compounds and their quantification ions (Fig. S15).





### 2.10 Pilot Mobile Field Measurements

Pilot mobile measurements were acquired July 20 to 21, 2022 in St. Louis, Missouri. MOIRA was deployed in a hybrid 2012 Toyota Prius V wagon vehicle and was powered by the vehicle's high voltage battery via an inverter and transformer (PlugOut v4.1 kit, PlugOut Power, LLC). A 47 mm PTFE filter was mounted toward the front and middle of the vehicle roof. The inlet tubing (1/8" diameter, 2.2 m length, PFA) was insulated together with a self-regulating heater, which maintained a temperature of 44 to 46 ºC during laboratory tests. The sample tubing heater was off during the July 20 measurements. Vehicle location was continuously recorded with a Qstarz BT-Q1000XT GPS datalogger, and the locations of sample collections are shown in Fig. S17.

This first mobile deployment uncovered several practical challenges which limited data recovery and time resolution, but which can be addressed with simple changes in future campaigns. First, the AC motor of the sample pump overheated, shutting down intermittently on July 20. An alternate pump was used on July 21, and mass flow meter measurements were used to correct all peak areas by normalizing by the ratio of the sample volume and the mean sample volume for each channel on July 21. Second, the barrel jack supplying power to the ethernet switch was sensitive to vibration on July 21. Since the computer had only a single ethernet port, only one RGA could be used to acquire data when this issue worsened. To address this, the power input has been soldered, and the ethernet switch has been mounted to the vibration-isolated portion of the instrument.

During these measurements, RGA A used a 2 ms scan method, while RGA B used a SIM method (Table S1). In the absence of calibration data, the sensitivity difference between the RGAs was corrected by multiplying all RGA B measurements by the slope of the line fitted to the peak areas of two samples which were consecutively collected at the same location (Fig. S18, slope = 3.17, $R^2$ = 0.99). One compound with Cook's distance exceeding four was excluded from this fit.

## 3 Results

### 3.1 Range of analytes

Example chromatograms analysed while cooking food during the indoor field study and from a liquid calibration standard injection are shown in Fig. 4. Table S3 includes a complete list of compounds identified from the indoor (n = 108), mobile (n = 40), and calibration measurements (n = 45). A wide range of compound classes has been detected:

- Alkanes: Straight-chain and branched alkanes with carbon numbers ranging from C5 to C15 are detected.
- Aromatics: The BTEX compounds (benzene, toluene, ethylbenzene, xylene) are all within MOIRA's range. Other aromatic compounds of note include 1,2,3-trimethylbenzene, 1,2,4-trimethylbenzene, and 2-methylfuran. Polycyclic aromatic hydrocarbons (PAHs) including naphthalene and 1-methylnaphthalene elute late in the chromatogram.





• Biogenic markers:  Isoprene, α-pinene, β-pinene, myrcene, and limonene, as well as isoprene's oxidation products, methacrolein and methyl vinyl ketone, were detected.

• Aldehydes:  Saturated aldehydes containing three to 11 carbons were detected during either the indoor or mobile measurements.  In addition, acrolein, furfural, benzaldehyde, and unsaturated aldehydes ranging from four to eight carbons were also detected in indoor samples.

• Alcohols:  Methanol is not well-retained by the trap and is thus used as the solvent in calibration standards. Alcohols ranging from two to eight carbons (including benzyl alcohol) were detected during the indoor field study, but breakthrough has not been evaluated for the smaller alcohols.  Based on retention time indices, it might be possible to detect larger alcohols including undecanol.

• Carboxylic acids:  Acetic, propanoic, and benzoic acids were all detected during the indoor field study.  In addition,
formic acid and larger acids such as hexanoic acid were detected during mobile measurements.

• Ketones:  Acetone and methyl ethyl ketone were detected during the field study with larger ketones (six carbons) detected in the calibration standard.

• Esters:  During the indoor field study, many ester and acetate compounds were detected, including several with similar mass spectra to 4-tert-butylcyclohexyl acetate.

• Sulfur-containing compounds:  During food cooking, diallyl sulphide and an unknown sulfur-containing compound were detected.

• Chlorine-containing compounds:  A different calibration mix from that shown in Fig. 4b included chlorinated compounds, including tetrachloroethylene (TCE).  1,1-dichloroethane and 1,2-dichloroethane were not well-retained by the trap as described in Sect. 3.4.

While retention time indices suggest that larger members of these series could be within the range of MOIRA as noted, it is possible that transfer in the inlet or instrument may limit this range, such that these species are not observed (Stein et al., 2009; Schauer et al., 2002).  In particular, the inlet tubing and quartz filter were not heated during the indoor measurements, and the inlet was not heated during the first day of mobile measurements.  In addition, breakthrough testing would need to be done for the most volatile compounds, for which collection may not be quantitative (see Sect. 3.4), though previous work
with the same collector and sorbent combination suggests safe sampling for compounds such as isoprene, methyl vinyl ketone, and methacrolein (Wernis et al., 2021).

Peak width is wider at the beginning of the chromatogram, decreasing from over 10 s full width at half maximum (FWHM) at pentane to less than 3 s for most compounds with retention times greater than 900 s (Fig. S10).  This suggests that more volatile early eluting analytes are not well-focused by the GC, such that their peaks are broadened by the time
distribution of their release from the trap during thermal desorption.



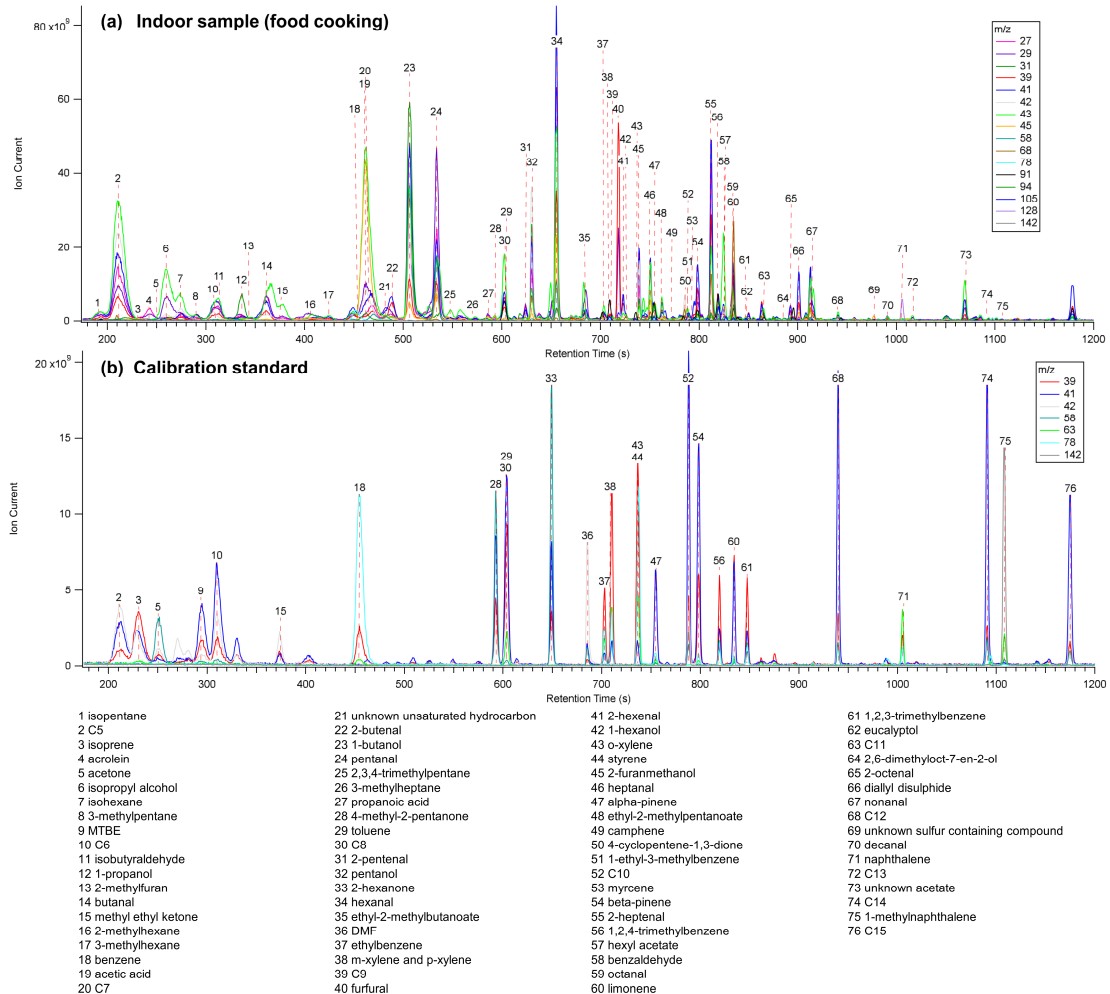

**Figure 4.** Single ion chromatograms from (a) an indoor sample during food cooking from the field deployment at a single-family residence and (b) analysis of a liquid calibration standard. Alkanes are listed as their carbon number.

### 3.2 Precision of GC temperature control and retention time

Consistency in retention time is important for comparison across channels and relies on precise trap and GC temperature control. For the indoor measurements, the range in each channel's GC temperature measurements at 1 Hz is typically less than 1 °C with greater variability occurring in the second GC ramp (orange) and hold above 180 °C (pink) (Fig. 5b). As a result of this precise temperature control within each channel, the standard deviation in retention time within each channel is



less than 2.5 s for the selected compounds shown in Fig. 5d with the exceptions of isoprene (channel B1) and tetradecane

(C14, channel A1).  However, the range between the channels is larger (up to 20 s at the beginning of the chromatogram), though this range is smaller (~10 s) and more consistent in the second half of the chromatogram.

While channels with consistently higher temperatures during the second GC ramp (orange shaded region) and hold (pink shaded region) tend to elute earlier as expected (Fig. 5c, Fig. 5d), interchannel deviations in temperature and retention time are not directly related during thermal desorption and the initial GC ramp.  This could be due to differences in insulation of

the GC hub and GC transfer lines that affect the uniformity of the GC column temperature, particularly when these temperature regions have a large difference in temperature setpoint at the beginning of the chromatogram.

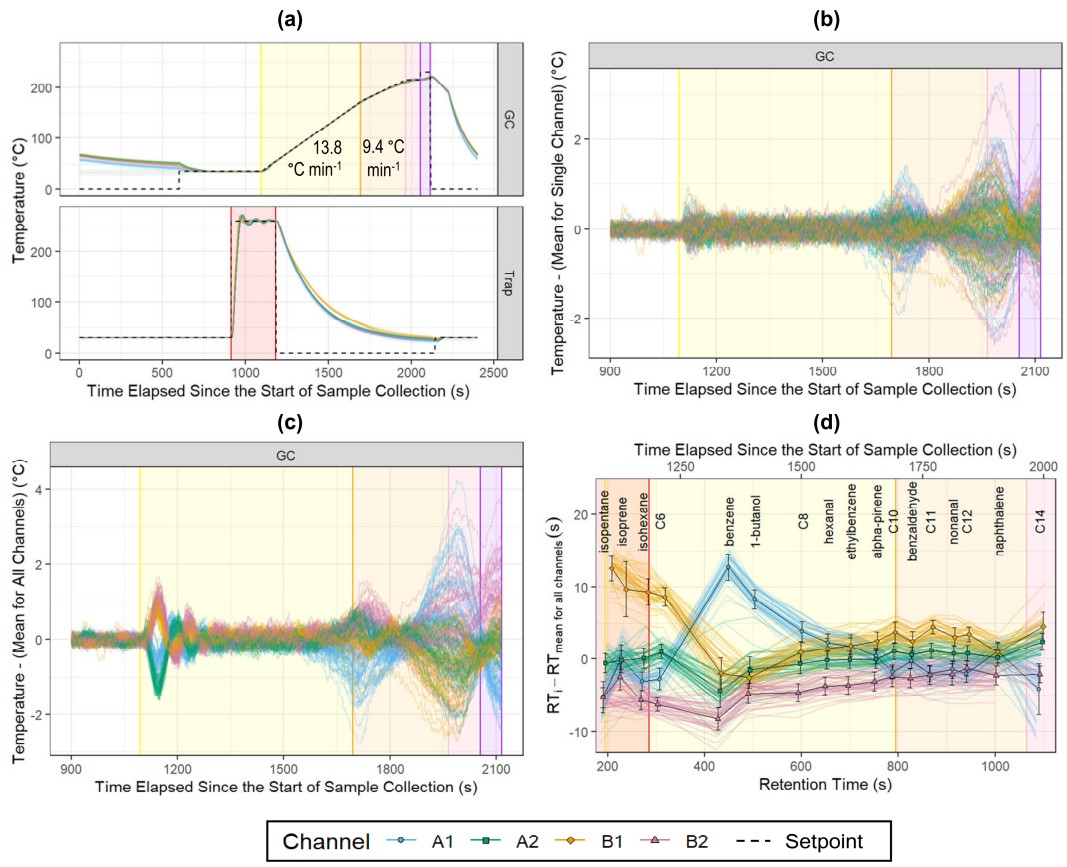

**Figure 5. Temperature and retention time data for the indoor measurements.  Colored lines represent individual samples ($n_{A1}$ = 52, $n_{A2}$ = 48, $n_{B1}$ = 47, $n_{B2}$ = 50), and black outlined points are channel means (error bars = ± 1 standard deviation).  (a) Measured**
**GC and trap temperatures and setpoints (1 Hz) (b) Repeatability of GC temperature control for each channel (1 Hz) (c) Deviation of GC temperatures from the mean temperature for all channels (1 Hz) (d) Deviation of peak retention times from the mean of a compound's retention time for all channels.  Alkanes are listed as their carbon number.**



### 3.3 Environmental conditions (ozone, relative humidity)

There are no significant compound-specific differences between collections at 10% and 95% relative humidity (Fig. 6).

Within channels, channel A1 tends to have lower recovery at 95% RH, and channel B2 recovery tends to be higher, but these interchannel differences are not significant.

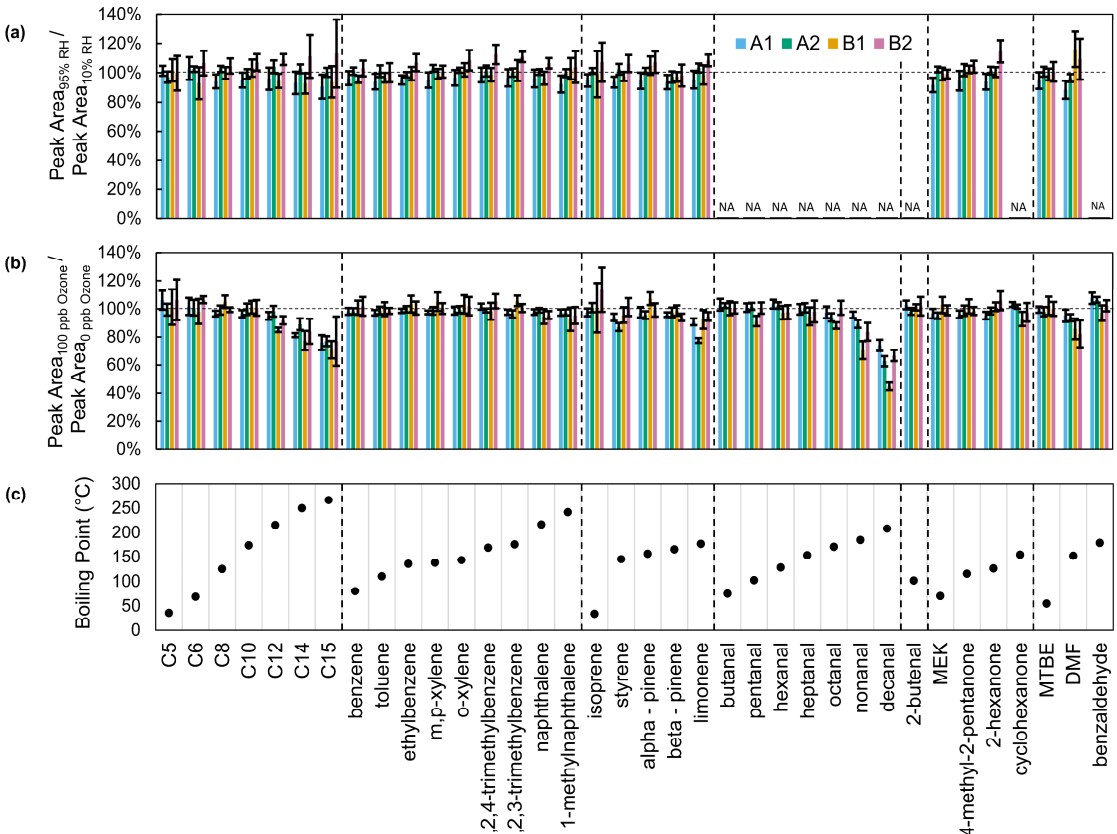

**Figure 6. Recovery of compounds in calibration standards with (a) 95% RH and (b) 100 ppb relative to zero air (10 % RH, 0 ppb ozone). (c) Boiling points of compounds in calibration standards. Alkanes are listed as their carbon number. Aldehydes and**
**cyclohexanone were included in a separate calibration standard, which was tested only with ozone.**

Of the selected compounds in the calibration standards, only limonene and α-pinene have expected ozonolysis lifetimes

that are less than four hours (Table S2). With 100 ppb ozone, limonene is depleted in all channels (minimum recovery: 77 ±

2 % for A2; maximum: 94 ± 3 % for B2), but not as much as expected (73% recovery for a 10 min sampling period) given

its gas-phase lifetime (τ = 32 min) (Fig. 6b). α-pinene recovery (minimum: 95 ± 3 % for A2; maximum: 107 ± 5 % for B1)

also exceeds expected recovery (88%) given its lifetime (τ = 80 min). Sorbent adsorption, the liquid standard injection



process, and partial ozone scrubbing by metal surfaces in the instrument may be mitigating ozone depletion. With the same collector and ozone concentration, Wernis (2021) observed a greater depletion of limonene (70% recovery) than observed in this work, likely due to longer sampling period (23 min vs. 10 min).

Unexpectedly, ozone decreases the recovery of larger members of the aldehyde and alkane series as well as for DMF
(Fig. 6). The long lifetimes of these compounds suggest that reaction is not the cause of this depletion (Table S2). Since this depletion covaries with compound boiling point, we hypothesize that ozone activates a surface which adsorbs these compounds. However, we are unsure as to why similar depletion is not seen for other less volatile compounds such as naphthalene and 1-methylnaphthalene.

An ozone scrubber could be placed at the instrument inlet to mitigate reaction with ozone. However, its location
upstream of the calibration standard injection ports would prevent use of liquid standards to characterize scrubber effectiveness and artifacts. Thus, in the absence of a system for gas calibration standards, we did not evaluate the effectiveness or artifacts of an ozone scrubber. We did not use an ozone scrubber for the mobile measurements (ambient ozone hourly average ranging from 31 to 64 ppb) or the wintertime indoor measurements. In future measurements, the inlet particulate filter could be replaced by quartz filters impregnated with sodium thiosulfate as done in other sorbent-based
offline and *in situ* VOC measurement methods (Helmig and Vierling, 1995; Wernis et al., 2021; Fick et al., 2001).

### 3.4 Analyte transfer and breakthrough

Analyte transfer and breakthrough were evaluated by varying the time between liquid standard injection and thermal desorption (Fig. 7). Pentane is omitted from this analysis, since the instrument response decreased over time for this compound, suggesting evaporation from the calibration standard. In addition, carboxylic acids are not represented in this test
since the standard solvent (methanol) was unsuitable.

Transfer from the injection port to the inlet is quantitative, as increasing sampling time does not result in increasing recovery of less volatile analytes (e.g. dodecane, naphthalene, 1-methylnaphthalee, tetradecane). In addition, most volatile analytes (including isoprene, hexane, and benzene) are well-retained by the sorbent collector with a 700 mL sample volume, which is equivalent to the 10 min sampling time used in this work. Exceptions are 1,2-dichloroethane (channel B2 only) and
1,1-dichloroethane (all channels), which both exhibit declines in response from 5 to 10 min sampling time.

The slight decrease in peak area for isoprene on all channels at 15 min (mean of 7% decrease relative to 10 min response) could indicate that this analyte might break through the trap if the method was altered to increase the sample time or flowrate above 10 min. However, the sorbent trap design as well as sorbent types and amounts were determined by Wernis (2021) to have breakthrough volumes greater than 2000 mL for isoprene as well as the sum of methyl ether ketone
and methacrolein (Wernis et al., 2021). As measured by the mass flow meters with a two-fold correction for gas type (30 mL min$^{-1}$), the five min helium purge following MOIRA's sample collection is approximately 150 mL. Thus, the 1200 mL total volume for a 15 min collection is still less than the reported breakthrough volume for these compounds. One possible



cause for this discrepancy is that our tests overestimate breakthrough since the liquid standard injection delivers all the analyte at the beginning of the sampling period (as opposed to continuously throughout).

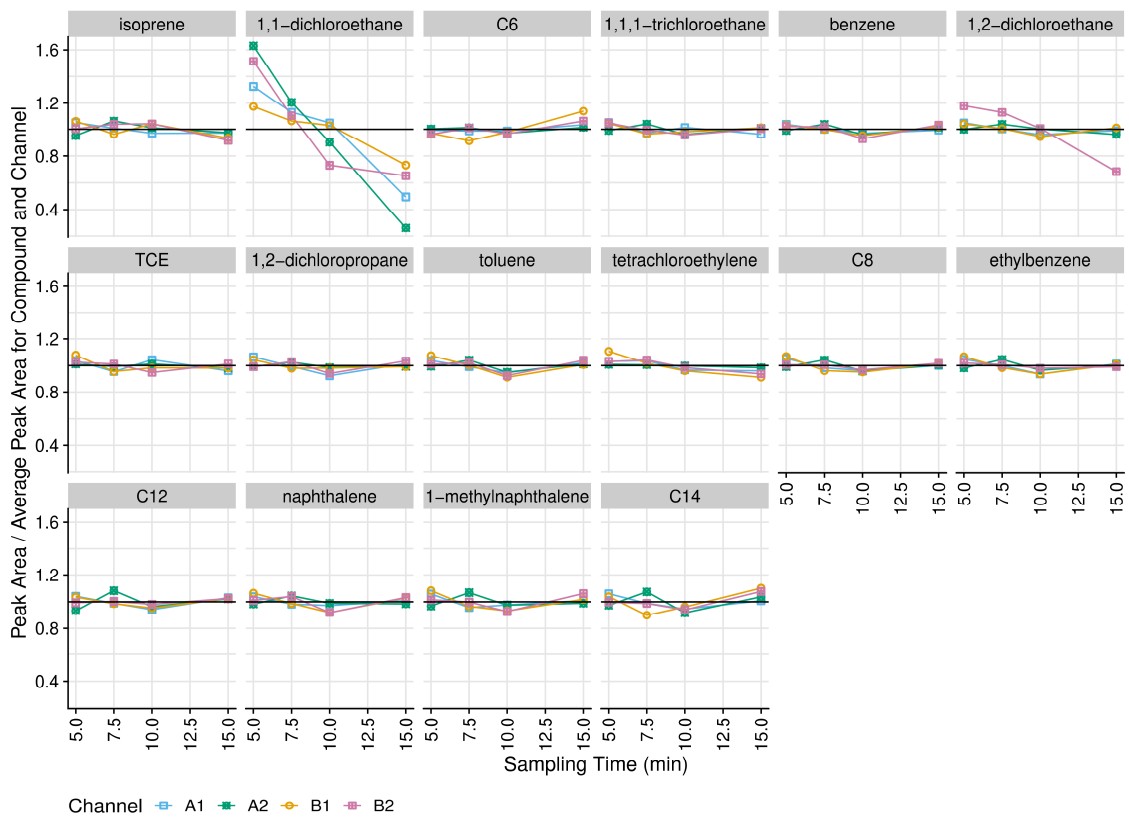

**Figure 7. Instrument response (normalized to mean for channel and compound) with respect to sampling time (70 mL min$^{-1}$ sample flowrate). Alkanes are listed as their carbon number.**

### 3.5 Laboratory calibration

Over the course of the experiment, the peak areas for the most volatile analytes (isoprene, pentane, methyl tert-butyl ether (MTBE), hexane) declined throughout the calibration curve experiment. We hypothesize that this is due to evaporation of the most volatile analytes from the liquid calibration standards. The calibration curve experiment started 21 days after the liquid calibration standard was prepared and ended nine days later after six replicates of the seven-point calibration curve were obtained for each channel. These compounds have thus been omitted from calculations of the precision of the instrument. In addition, sets of replicates for a given compound, channel, and concentration level for which less than six non-zero responses were obtained have been excluded from the calibration curve and precision calculations.



### 3.5.1 Minimum detected concentrations, limits of detection, and limits of quantification

The more sensitive RGA A is capable of measurements below 100 ppt with the full scan (2 ms dwell time per amu) for all compounds which were tested below that threshold in the seven-point calibration (Fig. 8c). In contrast, the mean minimum concentration levels detected by channels B1 and B2 with the 2 ms full scan method were 289 and 283 ppt respectively. For the SIM method, the lowest of the tested concentration levels (< 100 ppt for all compounds) is detected by all channels. Thus, to measure concentrations as low as 100 ppt, the higher dwell times of the SIM method should be used by RGA B (Table S1). As such, during the mobile measurements in this work, the 2 ms full scan method is used by RGA A, while a SIM method is used by RGA B.

Limits of detection (LOD) were calculated from the baseline noise of a blank chromatogram according to Eq. (2) (Fig. 8a):

$$LOD = \frac{3\sigma_{baseline}}{S} \tag{2}$$

where $\sigma_{baseline}$ is the standard deviation of the quantification ion at the compound retention time, and S is the slope of the calibration curve. For each channel, MS baseline noise as a function of retention time is calculated as the rolling standard deviation of the signal of the quantification ion from a blank chromatogram. The windows for standard deviation calculations are centered and contained 21 scans. To calculate $\sigma_{baseline}$, the standard deviation is smoothed with a two-pass, boxcar algorithm whose window width was 250 scans, which is one-tenth of the total number of scans in the chromatogram.

The baseline noise for RGA B is higher than that for RGA A, reflected in the higher LODs for channels B1 and B2 in Fig. 8a. For both RGAs, the LODs calculated by Eq. (2) (Fig. 8a) were less than the minimum concentrations levels detected in the seven-point calibration curve (Fig. 8c), indicating that the MOIRA instrument is not currently as sensitive as predicted by Eq. (2).

Finally, the limit of quantification (LOQ) was calculated from the linear calibration curves (Fig. 8b):

$$LOQ = \frac{10\sigma_{intercept}}{S} \tag{3}$$

where $\sigma_{intercept}$ is the standard error of the intercept of the calibration curve. In comparison to LOD and the minimum detected concentration, the limits of quantification for RGA A were similar to those for RGA B. For the 2 ms full scan method, mean LOQ ranged from 311 ppt for A1 to 562 ppt for B1; for the SIM method, the mean LOQ ranged from 41 ppt for A1, A2, and B2 to 42 ppt for B1.



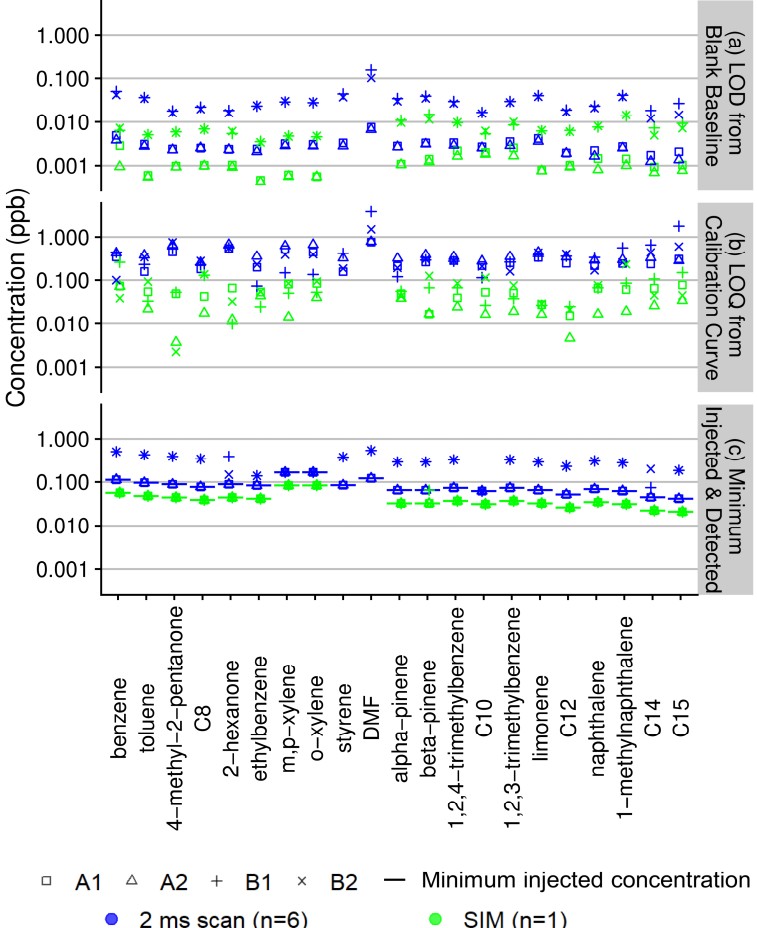

**Figure 8.** (a) Limit of detection (LOD) calculated from the baseline of a blank chromatogram (Eq. (2)) (b) Limit of quantification
(LOQ) calculated from the calibration curve (Eq. (3)) (c) Minimum concentration detected among calibration samples (horizontal
lines indicate the minimum injected concentration in the calibration experiment). Due to significant evaporation from the
calibration standard, the most volatile analytes (e.g. isoprene, pentane, hexane) are not included here. Alkanes are listed by their
carbon number.

### 3.5.2 Evaluating correction methods for interchannel sensitivity differences and instrument drift over time

As described in Sect. 2.7, a daily correction factor compensated for the drift in instrument sensitivity throughout the
multi-day calibration experiment. Only a single concentration level of the seven was used to calculate this daily correction
factor to simulate the use of a daily tracking standard during a field or lab campaign, as is done in the indoor measurements
in this work. The Pearson $R^2$ for the daily correction factors for A1 and A2 is 0.94, suggesting that drift in the shared MS



detector was the main contributor to changes in sensitivity over time, though the correlation for B1 and B2 was not as strong

($R^2$ = 0.45). To evaluate if the daily correction improves measurement repeatability, the normalized root mean square

deviation (NRMSD) from the mean for each channel was calculated (Fig. S13):

$$NRMSD = \sqrt{\frac{\sum\left(\frac{y-y_{mean}}{y_{mean}}\right)^2}{N}} \qquad (4)$$

where y is the instrument response, $y_{mean}$ is the mean response for the same compound, channel, and concentration level,

and N is the total number of measurements. When the daily correction factor was used, the NRMSD improved for all

channels: 9 to 6% for A1, 7 to 4% for A2, 15 to 14 % for B1, and 12 to 11 % for B2. These NRMSD values represent the

single-channel repeatability or precision.

The linear calibration curves demonstrate good linearity, though a power fit is more representative at lower

concentrations for some compounds (Fig. 9, raw peak areas shown in Fig. S11, Pearson $R^2$ for other compounds in Fig. S12).

While the channels that share an RGA have similar sensitivity, RGA A is approximately seven times more sensitive than

RGA B in this calibration experiment. To show the four channels on a similar scale, the data in Fig. 9 is corrected with a

channel sensitivity factor F which is the median ratio (across all compounds) of the calibration curve slope $m_{i,Channel}$ for a

given compound i and channel and the mean slope for that compound for all four channels:

$$F_{Channel} = median\left(\frac{m_{i,Channel}}{mean(m_{i,A1},\ m_{i,A2},\ m_{i,B1},\ m_{i,B2})}\right) \qquad (5)$$

Thus, the deviations between the calibration curves for the different channels in Fig. 9 are proportional to the errors if the

channel sensitivity factors were used to compensate for the RGA sensitivity difference instead of a calibration curve. For

example, channels A1 and A2 have a slightly higher sensitivity to more polar (e.g., dimethylformamide) and less volatile

(e.g., naphthalene) compounds. This may indicate an active site between the calibration standard injection tee shared by B1

and B2 and the rotary valve.

Since the RGA sensitivity difference is larger than other interchannel differences, a similar RGA sensitivity factor can

be calculated:

$$F_{RGA} = median\left(\frac{mean(m_{i,A1},\ m_{i,A2})}{mean(m_{i,A1},\ m_{i,A2},\ m_{i,B1},\ m_{i,B2}))}\right) \qquad (6)$$

Both the channel sensitivity factors and RGA sensitivity factors can be used to correct for the sensitivity differences of the

four channels, particularly when a compound is not present in a calibration standard:

$$For\ all\ channels:\ Peak\ Area_{Corrected} = Peak\ Area\ /\ F_{Channel} \qquad (7)$$

$$For\ RGA\ B\ only:\ Peak\ Area_{Corrected} = Peak\ Area * F_{RGA} \qquad (8)$$




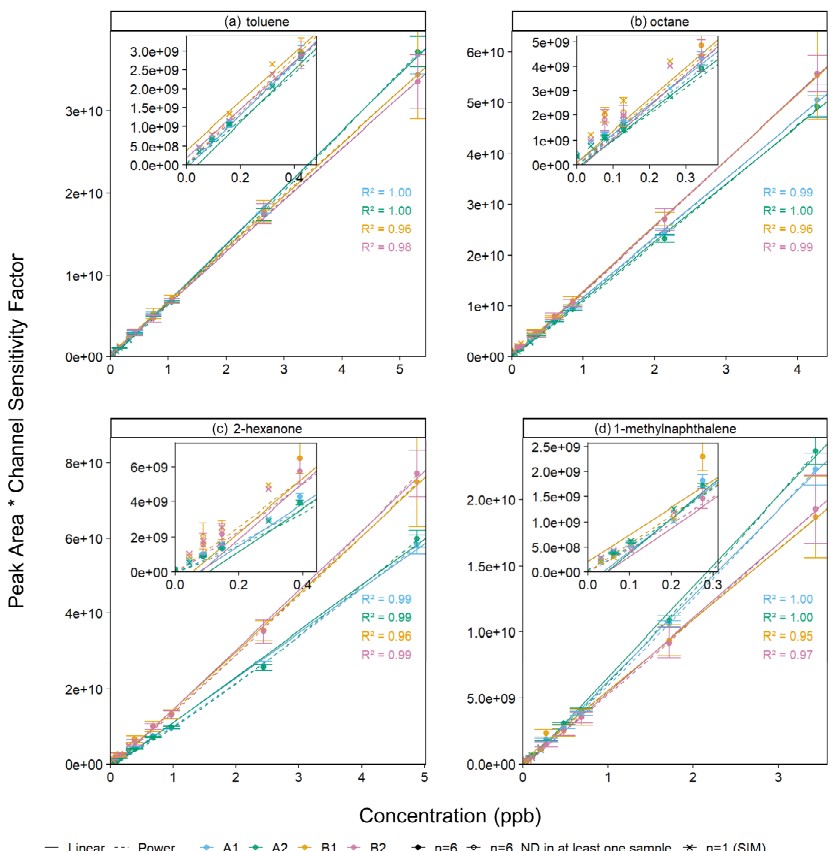

**Figure 9. Linear and power calibration curves (error bars ± 1 standard deviation) for four example compounds. For similar scale across channels, peak areas are corrected by each channel's sensitivity factor ($F_{A1}$ = 1.68, $F_{A2}$ = 1.83, $F_{B1}$ = 0.227, $F_{B2}$ =0.250), which is the median ratio (across all compounds) of the calibration curve slope for a given compound and channel and the mean slope for that compound for all four channels. The inset plot shows the lower range of data. Open circles indicate concentration levels for which no peak was detected on among at least one of the 2 ms scan replicates (n=6). Pearson $R^2$ values include only replicates for which n=6 (whose mean is represented by the filled circles).**

To compare these two approaches with linear and power law calibrations, the repeatability of the instrument's 4-channel response was evaluated with the NRMSD with respect to the mean response $y_{mean}$ across all channels for the same compound and concentration level (Fig. S14). For all compounds in the calibration standard, the NRMSD for the channel and RGA sensitivity factor methods (14% and 14%, respectively) are only slightly higher than that of the linear calibration curve (13%) and better than that of the power law calibration curve (16%). In comparison to the four-channel repeatability, the single channel repeatability (calculated as the NRMSD with respect to the mean response for a single given channel) is much





better for channels A1 (6%), A2 (4%), and B1 (11%), but similar for channel B2 (14%). While the interchannel repeatability of the power law calibration curve was worse than that of the linear calibration curve, the power law better fit the SIM data of some compounds at lower concentrations, which are not included in the regression. When the 4-channel precision is calculated for individual compounds, the NRMSD is highest for dimethyl formamide, tetradecane, and pentadecane, suggesting interchannel differences in the transfer of these more polar or less volatile compounds (Fig. S14).

**3.6 Indoor measurements**

Fig. 10 shows a timeseries of eight compounds from nine hours of the indoor measurements, and the last column in Table S3 lists all compounds which were quantified in this pilot study. Not all of these compounds are in the calibration standard, so the relative abundance with the compound-specific correction described in Sect. 2.9 is shown (normalized to the maximum value) instead of the calibrated mixing ratio. This compound-specific correction leverages timeseries dependence in order to

merge the RGA A and RGA B data into a single timeseries without resorting to smoothing, which would distort sudden changes in concentration. For this same time period and selection of compounds, Fig. S16 also includes raw peak areas, mixing ratios from the power law calibration curve for compounds in the calibration standard, and responses that have been corrected with the channel factor approach described in Sect. 3.5.2. The compound-specific factor provides a more precise correction than the channel sensitivity factor, perhaps because of the more diverse range of compounds detected relative to

that of the calibration curve experiment.

The activities in this period (which are numbered in Fig. 10) include using a plug-in air freshener (1), stir-frying vegetables (2-5, 12-14), washing dishes by hand (6), cleaning the floors with a pine scented cleaner (7, 17), and window opening (11-12, 15-16). Before the air freshener is plugged in (1), hexyl acetate is only detected by the more sensitive A1 and A2 channels. Hexyl acetate increases in concentration when the air freshener is plugged in (1) and decreases when

ventilation increases (opening windows from 11 to 12, opening windows and turning on vent fan from 15 to 16). Limonene and α-pinene increase during both stir-frying periods with timing dependent on the addition of black pepper (last step (5) in the first stir-fry, first step (12) in the second stir-fry). Of the compounds shown, others with large enhancements during cooking include octane and isobutyraldehyde. During the first stir-fry, the early increase in camphene may be from emissions when the vegetables were chopped (2). After dishwashing (6), limonene abundance increases. When cleaning the

floors with pine-scented cleaner (7, 17), camphene abundance increases, but the already abundant concentration of limonene after stir-frying likely masks the expected emissions of this compound from the pine-scented cleaner. When the door between the kitchen and the house's lower level is opened (6 to 7, 9 to 10, 17), naphthalene and toluene abundances increase, possibly due to airflow from the lower level to the kitchen. The house's lower level includes a single-car garage, where a car is regularly parked and a gas-powered push lawnmower, gasoline container, and household chemicals including mothballs

are stored.





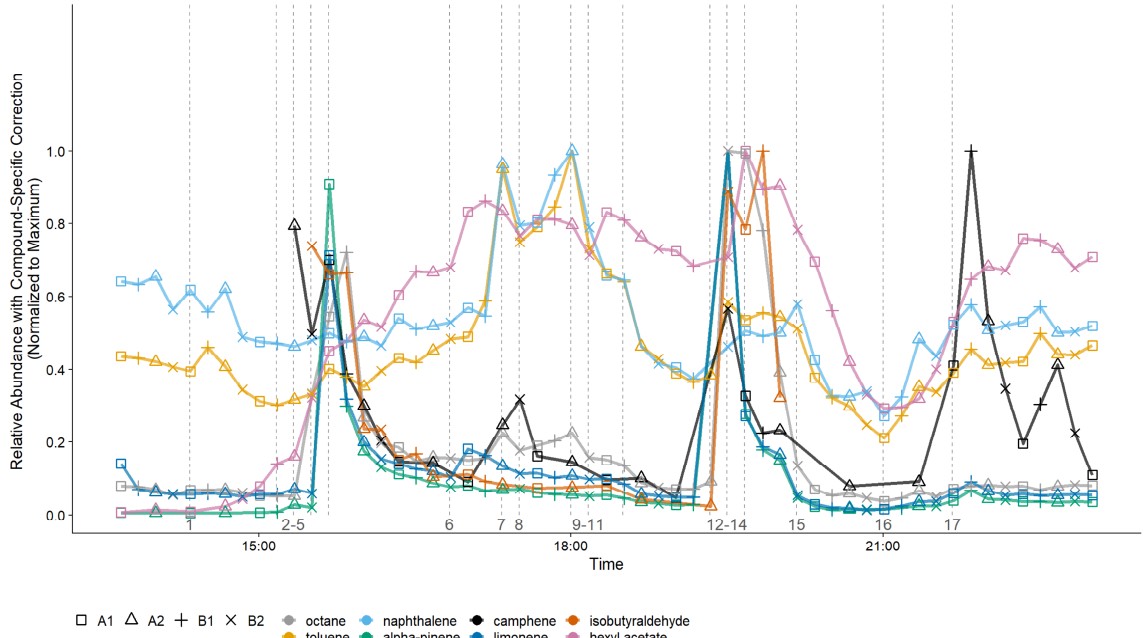

**Figure 10. Timeseries of relative abundance with compound-specific correction and normalization to the maximum measurement. The vertical lines correspond to the following activities (with times rounded down to the nearest sampling time): 1 - Air freshener plugged in, 2 - Chopped vegetables, 3 - Heated vegetable oil, 4 - Stir-fried onion, garlic, and ginger, 5 - Stir-fried mushroom with black pepper, 6 - Washed dishes, 7 - Cleaned floors with pine cleaner and opened kitchen door, 8 - Closed kitchen door, 9 - Opened kitchen door, 10 - Closed kitchen door, 11 - Opened windows, 12 - Closed windows, heated oil and black pepper, 13 - Stir-fried onion, 14 - Stir-fried mushrooms, 15 - Opened windows, turned on vent fan, 16 - Closed windows, turned off vent fan, 17 - Cleaned floors with pine cleaner, opened kitchen door.**

**3.7 Spatially-resolved concentration mapping from pilot mobile measurements**

The pilot mobile measurements in this work are the first attempted with the MOIRA instrument. Forty compounds were quantified, and maps of toluene and methyl ether ketone concentrations are shown in Fig. 11. As described in Sect. 2.10, this initial work uncovered now-resolved issues with the sample pump and ethernet switch that limited data recovery. If data recovery had been complete, the samples in Fig. 11 would have required less than four hours of sampling time.

Previous mobile measurements of toluene have been associated with petroleum fuel emissions including by vehicles, and in this work, toluene concentrations were highest in samples during which the mobile platform was parked at gasoline and diesel fueling stations (Gkatzelis et al., 2021; Healy et al., 2022). Methyl ethyl ketone (MEK) has been previously apportioned to volatile chemical products and chemical waste facilities, and the sample with the highest MEK concentration was collected nearby a volatile chemical products vendor (Gkatzelis et al., 2021; Healy et al., 2022). Other notable



compounds include isoprene and its oxidation products, including methacrolein, methyl vinyl ketone, formic acid, and acetic
acid (Table S3, last column identifies compounds observed during mobile measurement).

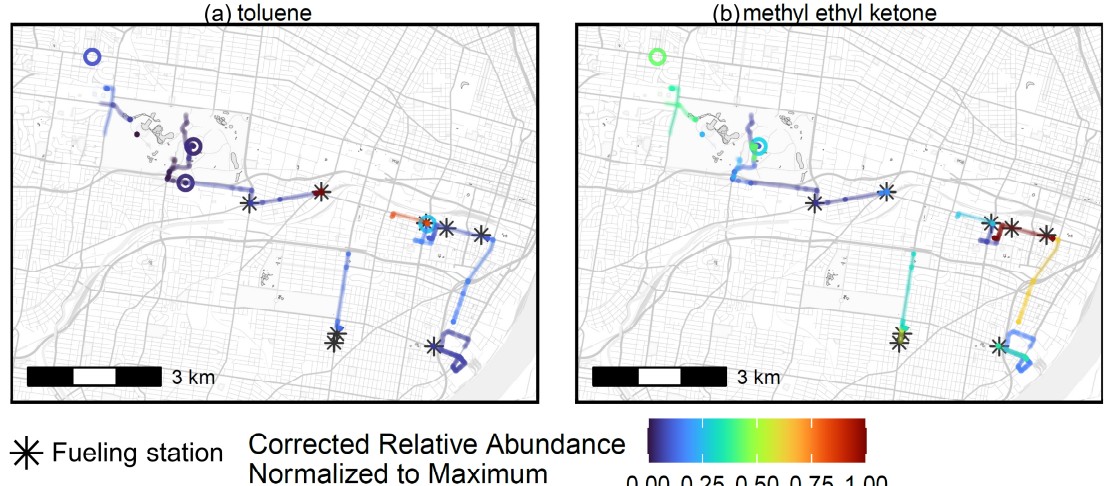

© OpenStreetMap contributors 2023. Distributed under the Open Data Commons Open Database License (ODbL) v1.0.

**Figure 11.   Map of normalized relative abundances of toluene and methyl ethyl ketone measured during pilot mobile measurements in St. Louis, Missouri (July 20-21, 2022). Gasoline and diesel fueling stations within 50 m of sampling locations are indicated . Stationary collections are represented by open circles, while samples collected while the vehicle is moving are**
**represented as lines.**

## 4   Discussion

MOIRA integrates four TD-GC MS measurement channels in a single instrument, enabling faster time resolution with synergies in footprint, power optimization, and cost from shared control, electronics, and vacuum systems. Continuous 10 min integrated samples have fast enough time resolution to capture dynamic changes in the indoor environment or to map
VOC concentrations with mobile sampling, while achieving sufficient GC performance and measurement sensitivity. The instrument's first mobile measurements described in this work demonstrate the feasibility of using MOIRA for spatially resolved measurements from a hybrid vehicle, using its high voltage battery to power the instrument. While vibration and power challenges limited data recovery in this first mobile deployment, these issues do not involve the GC or MS sections of the instrument and have been addressed in preparation for future measurements. With a volatility range bounded by pentane
and pentadecane, compounds measured include aliphatic and aromatic hydrocarbons as well as oxidized species including carboxylic acids, ketones, and aldehydes. Key markers of biogenic emissions (e.g. isoprene, monoterpenes), atmospheric oxidation (e.g. methacrolein, methyl vinyl ketone, formic acid), and anthropogenic emissions (e.g. methyl ethyl ketone, BTEX (benzene, toluene, ethylbenzene, xylene), naphthalene) are detected.



The compact footprint of the MOIRA system, which fits in a hybrid Prius wagon vehicle, is enabled by the miniature
GC heaters, which each accommodate a standard 30 m column, and the compact RGA detectors. Selecting RGAs as
detectors strikes a balance of capability, cost, and footprint. The cost of two RGAs is an order of magnitude less than that of
a high-resolution time-of-flight MS detector typical of PTR-MS and other Chemical Ionization-MS systems. In this work,
we demonstrate that this sensitivity is sufficient to detect markers in ambient outdoor air as well as those for indoor activities
such as cooking, cleaning, air freshener use, and ventilation changes. In contrast to soft ionization MS methods or the less
expensive FID detector, the selectivity of EI MS enables non-target identification as well as differentiation of co-eluting
compounds (e.g., toluene and octane), which is especially important with the shorter 20 min GC program. In addition, the
physical footprint of two RGAs and their shared vacuum system is close to that of a single typical benchtop quadrupole MS
detector, enabling deployment in a wider range of vehicles.

The multichannel design of the MOIRA system is well suited for combined studies of indoor and outdoor air, as it is
capable of simultaneous sampling from multiple inlets. Additionally, different collectors and GC column materials could be
simultaneously deployed in different channels to extend the range of target analytes. Using multiple instances of the same
collector and GC analysis components, as demonstrated in this work, provides opportunities to evaluate the consistency and
performance of these components, many of which are custom.

### 4.1 Considerations for volatility range and time resolution

The measurement method described in this paper has a time resolution of 10 min and was intended to collect the widest
volatility range (pentane to pentadecane) of analytes while still maintaining high precision in GC temperature control for
consistent retention times. The smaller peak widths at higher retention times could accommodate a faster GC ramp at higher
temperatures. However, faster ramp rates at temperatures above 170 °C resulted in less precise, less accurate temperature
control. If the maximum GC temperature was further increased (from 230 °C up to 280 °C maximum for the column phase)
to expand the volatility range, either the minimum temperature would need to be increased, compromising the resolution of
the most volatile analytes, or more interchannel variance in retention time would need to be tolerated. Alternatively, the time
between measurements could be increased to accommodate a longer GC program. In either case, the transfer lines would not
limit the volatility range, as their individually controlled setpoints could be increased, though power requirements would also
increase.

Increasing the analyte range to include more volatile compounds would require better focusing at the head of the GC
column and possibly at the sorbent trap. However, decreasing the initial GC temperature below 35 °C and the trap collection
temperature below 30 °C would require increasing instrument complexity with either cryogenic or thermoelectric cooling as
well as a more complicated approach for addressing sample humidity. In the current design, a higher trap temperature
during collection (30 °C), use of hydrophobic sorbents, and a helium purge between collection and thermal desorption have
been sufficient to prevent large amounts of water from entering the GCs or vacuum chamber.



Faster time resolution could be achieved if a smaller range of analytes was targeted with a narrower GC temperature range. Since sampling volume is proportional to sampling time, the limit of detection (expressed as a mixing ratio) would increase with a shorter sample collection. This loss of sensitivity could be mitigated by increasing the mass spectrometer dwell time with selected ion monitoring or by overlapping sample collections. In the method described here, each sample

collection is followed by a 5 min helium purge to remove water from the sorbent collector, but tests indicate that purging time could be decreased to as low as 3 minutes at 28 kPa to either shorten or extend and overlap sample collection. However, a longer sampling time and volume might increase artifacts from breakthrough and ozone interferences.

### 4.2 Limitations

Species detected by the more sensitive A1 and A2 channels may be below detection for the B1 and B2 channels when using

a full scan MS method (e.g., indoor measurements of hexyl acetate prior to turning on air freshener in Fig. 10). Fortunately, sufficient multichannel sensitivity for ambient measurements can be achieved by using SIM methods for RGA B measurements and full scan methods for RGA A, though this reduces the full-scan data available for non-target identification.

In addition, the sensitivity difference between the RGAs necessitates a correction to merge data from the four channels

into a single dataset. The calibration curve data in this work shows that there is some loss of precision in merging the four-channel data into a single dataset, whether a calibration curve, channel sensitivity factor, or RGA sensitivity factor is used. In the indoor and mobile measurements, the correction for RGA sensitivity was achieved with two alternate approaches that use the measurements themselves to identify appropriate correction factors. While the indoor dataset demonstrates the benefits of calculating correction factors by compound, the success of the RGA and channel sensitivity factors in the

calibration curve experiment suggests that simpler approaches, such as the regression of responses of two similar samples in the mobile measurements, can also be valid. The most appropriate approach will vary by dataset, dependent on the calibration data available as well as the similarity of the measured compounds to those in the calibration standard.

Using a limited set of internal standards would aid in better characterizing the RGA sensitivity difference as well as drift in detector response over time. In particular, a system for gas calibration standards could improve quantification of the most

volatile compounds, since analytes such as isoprene, pentane, MTBE, and hexane were found to evaporate over time from liquid standards during storage in both this work and that of others (Wernis et al., 2021). The constant delivery concentration would also enable more accurate measurement of breakthrough than with liquid standards, whose entire volume is delivered into the injection port at the beginning of a sampling period. Gas calibration standards would also enable better characterization of ozone interferences than the current set-up, which does not permit the introduction of a

scrubber downstream of the liquid standard injection port. Alternatively, a liquid calibration system would likely be more compact and would allow more flexible and economical introduction of different calibration mixes while providing improvement in quantification over the manual injection method used here (Wernis et al., 2021).

While easily identified, shadow peaks, or interferences between the two RGAs, could be mitigated by decreasing the size of the gaps between the vacuum chamber wall and the partition which separates the RGAs. In addition, an ozone
scrubber (such as a glass fibre filter impregnated by sodium thiosulfate) should be used in future measurements with elevated ozone. Finally, a heated sample line was used for only part of the mobile measurements and was not used in the indoor measurements, potentially compromising sample transfer of less volatile analytes.

## 5 Conclusion

MOIRA integrates multiple miniaturized GCs and compact residual gas analysers in a single instrument, occupying a
physical footprint and requiring power suitable for deployment in a hybrid hatchback wagon vehicle. Inherent in this multichannel approach is the necessity of characterizing the differences between channels. Nevertheless, the accompanying increase in time resolution enables efficient application of *in situ* GC-MS in mobile deployments. MOIRA's 10 min time resolution is also sufficient to measure dynamic changes in indoor VOC concentrations due to many household activities with a lower cost than that of other *in situ* VOC instruments such as the PTR-MS. In addition, the combination of GC
separation and electron ionization provides selectivity for both target and non-target compounds. In future work, the MOIRA system could be flexibly adapted for other multichannel approaches by substituting alternate thermal desorption collectors, sorbents, or column phases in one or more channels.

### Data Availability

Data is available upon request to the corresponding author (Brent Williams, brentw@wustl.edu).

### Author Contribution

BW and JT conceptualized the instrument objectives, approach, and field testing plan. AD, BW, and NK designed the instrument. AD, SH, and RH constructed the instrument. AD developed instrument operation software and data analysis software. AD performed lab testing of the instrument. AD, TC, JC, and BW carried out field measurements. AD prepared the manuscript with contributions from all co-authors.

### Competing Interests

The authors declare that they have no conflict of interest.



**Disclaimer**

Any opinions, findings, and conclusions or recommendations expressed in this material are those of the author(s) and do not necessarily reflect the views of the National Science Foundation.

**Acknowledgements**

We are grateful to J. Linders and J. Graflage of the WUSTL Chemistry Department Machine Shop for their advice and skill in the design and fabrication of key vacuum system components. Thanks to R. Wernis for advice on the use of the sorbent traps. This material is based upon work supported by the National Science Foundation Graduate Research Fellowship under DGE-1745038 and DGE-2139839 and by the National Institute of Health under grant P42-ES023716 and T32-ES011564.

Indoor air test measurements were additionally supported by the Alfred P. Sloan Foundation under grant 2018-11133.

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
