# Peer review of "Development of a Multichannel Organics *In situ* enviRonmental Analyzer (MOIRA) for mobile measurements of volatile organic compounds"

_EGUsphere, 2023_

## Author Response (AR1)

The authors thank both reviewers for their thoughtful and encouraging comments. Below are responses to the reviewer comments (in bold) with new or edited text in italics. References as well as revised tables and figures are provided at the end of these responses.

**RC1: 'Comment on egusphere-2023-1479', Anonymous Referee #3, 08 Jan 2024**
**This paper falls within the scope of AMT and should be published with minor revisions. The authors present a novel instrument of a 4 channel GCMS that can fit in the back of a Prius. The authors clearly lay out the instrument information as well as key calibration work to test the validity of the instrument and ability to use all 4 channels together. The paper is laid out well and shows how this new instrument fits in with a history of mobile instruments and what instrumental gap it fills.**

**Below are some minor comments or points of confusion for this reviewer:**

**General comments:**

**The use of the "Prius wagon vehicle" is awkward language. Is there a better way to re-phrase this? Maybe some combination of a Prius wagon, a hybrid vehicle?**

> Following this suggestion, we have adopted "hybrid vehicle" throughout the manuscript, specifying in the methods (Section 2.10) that the vehicle is a "2012 Toyota Prius V."

**The ordering of figures and tables in the supplement doesn't match the main text. It also doesn't seem specifically done in relation to text in the supplement either.**

> The supplement has been organized in thematic sections, grouping figures and tables of related subject matter: S1 Instrument design, S2 Shadow peaks, S3 Relative quadrupole transmission efficiency, S4 Selective ion monitoring methods, S5 Chromatogram characteristics and quantification, S6 Indoor measurements, and S7 Mobile measurements.

**Section specific:**

**Lines 593-597: Why did you not try overlapping sampling? You mention they are non-overlapping and that (line 593) "This loss of sensitivity could be mitigated by increasing the mass spectrometer dwell time with selected ion monitoring or by overlapping sample collections" with only (line 597) "However, a longer sampling time and volume might increase artifacts from breakthrough and ozone interferences." listed as to why you didn't try this. Was it tried but not shared? Where does this idea that there could be more interferences come from? Wouldn't that have been a good way to check the different channel sensitivities? Especially that the different correction factors were valid?**

> With the multichannel design, there is actually only two additional min of possible overlapping sample time per sample, since our tests indicate that at least three min of purging are required after sampling to remove water. In the method schematic in Fig. 3d, this is equivalent to the extension of the green sampling bar by 2 min and the shortening of the blue purge bar by the same duration. This could not facilitate a direct comparison of the channels, whose samples would not be completely overlapping and would also result in an uneven sample flowrate, which could complicate experimental design when other instruments shared an inlet. Increased sample time could cause breakthrough (given larger sample volume) for the most volatile analytes (Section 3.4) as well as increased ozone artifacts with longer exposure to ambient levels (Section 3.3). Indeed, the short 10 min sampling time may be one cause of the less severe ozone artifacts observed with MOIRA in comparison with other uses of this sampling trap (Wernis et al., 2021) (Section 3.3). Finally, while not all ions are measured, a SIM method can improve sensitivity by orders of magnitude, as opposed to a 20% increase in analyte loading with the slightly longer sample collection (Section 3.5).

> To better reflect our appraisal of these compromises, the text has been changed with updates bolded here:
> *"In the method described here, each sample collection is followed by a 5 min helium purge to remove water from the sorbent collector, but tests indicate that purging time could be decreased to as low as 3 min at 28 kPa to extend and overlap sample collection **by 2 min. However, during this 2 min of overlap, the total flowrate at the instrument inlet would be doubled, potentially complicating experimental design. In**

*addition, the* longer sampling time and volume might increase artifacts from breakthrough and ozone interferences *(Section 3.3, Section 3.4)."*

**Line 265: do you mean you did the Cook's distance test multiple times until the threshold was met? Is that valid? "Measurements which had a Cook's distance of greater than four were removed, recalculating this parameter without the excluded data until all points met this threshold." The discussion here is unclear.**

Yes, that is correct. The test was applied multiple times. The underlying assumption of this approach is a constant rate of change of concentration. The multiple iterations of the Cook's distance test are used to remove the outsized influence of dynamic events which do not fulfill this assumption (e.g. the initial increase in concentration at the beginning of the stirfry), not to remove statistical outliers in a strict sense.

This text, which also includes a quantitative evaluation of the compound-specific factor as suggested by the other reviewer, has been added to the end of Section 2.9:

*"The underlying assumption of this approach is a constant rate of change of concentration over three samples (30 min). For example, if the species concentration increases at a constant rate, then the concentration of a given sample is equal to the average of the previous and subsequent sample. The multiple iterations of the Cook's distance test are used to remove the influence of the most dynamic events which do not fulfill this assumption (e.g. the initial increase in concentration at the beginning of cooking), not to remove statistical outliers in a strict sense. For compounds quantified with liquid calibration standards, the root mean square deviation of the compound-specific factor from the ratio of the means of the calibration curve slopes for RGA B channels and RGA A was 14%."*

**Figure 6 caption: 100 ppb Ozone relative to zero air, word ozone is missing**

Thank you, we have added the word ozone.

**Lines 396-397: The volume math and logic was difficult to follow as written. How did you get from 5 min @ 30 mL/min = 150 to 1200 mL in 15 min, was it the same 30 mL/min rate? Or something different?**

We have added the sample flowrate parameter during breakthrough tests: *Thus, the total sample and purge volume for a 15 min collection at 70 mL min$^{-1}$ is 1200 mL, which is still less than the reported breakthrough volume for these compounds.*

**Section 3.5.1 is hard to follow in parts, in particular the minimum injected concentration. Was that done for only A1 and A2? Because it was more sensitive? It was also hard to tell at first the horizontal line through the symbol is what is represented and not a separate line. It looks more like an error bar through the symbol. It doesn't seem like the 'minimum injected concentration' is referenced in the main body text. Perhaps that would help with any confusion.**

Thanks for helping us identify this opportunity for increased clarity. We included the minimum injected concentration for each compound in order to distinguish instances in which the instrument was sensitive to the lowest calibration curve level, such that the calibration curve experiment overestimates the limit of detection. The following text has been added: *"In Fig. 8c, the minimum detected concentrations in Fig. 8c are equal to the minimum injected concentrations (ie. the smallest level in the calibration curves) for all but one SIM measurement and all of the 2 ms full scan measurements by RGA A. Thus, the actual instrument sensitivity for these measurements is better than suggested by the minimum detected concentrations shown in Fig. 8c."*

We have also made several changes to Figure 8 (included at the end of these comments). As this reviewer points out, the horizontal lines for minimum injected concentration are especially confusing as they resemble error bars. We have removed these lines and instead have added an additional data series (filled circles). In addition, we have made several other changes to increase clarity (e.g. individually-scaled, more granular y-axis gridlines; spacing out the markers from different channels).

**Figure 9: Formatting looks odd but probably not the author's fault, larger font on the legend would be nice**

> The font size of the legend has been enlarged for better readability among other minor changes for clarity, and we have prepared a vector graphics version of the figure as per the second reviewer's comments. The revised figure is included at the end of these responses.

**Section 3.7: How fast was the Prius going? What distance was covered per sample? An example for context would be nice.**

> We have selected an example for context as suggested, labeled it in Figure 11 (see end of these responses), and added the following to the main text: *"As an example, during the collection of the sample labelled in Fig. 11a, the vehicle travelled 2.2 km at an average velocity of 13 km hr⁻¹."*

> The revised figure is included at the end of these responses. The caption has been edited to add the following: *"As an example, the total distance of one of the mobile samples (2.2 km) is indicated in (a)."*
* * *
**RC2: 'Comment on egusphere-2023-1479', Anonymous Referee #1, 21 Jan 2024**
**This manuscript describes a new compact, multi-channel gas chromatograph coupled to mass spectrometer designed to provide continuous data sampling with significantly shorter sampling periods (typical 10 min) the customary 1-hour with FID-based AutoGC systems. Since the system is also highly mobile and relatively low-power, it will fill a need for high-speciated and quantitated measurements of ambient air in urban and indoor environments. Authors demonstrate superior technical and analytical skill with the material presented. The complexity of the instrument leads to considerable work required to resolve differences between different channels, which the authors present in detail. This is a good manuscript and should be published after minor revisions to address the issues presented below.**

**General comments**

**Line 166. "the two signals is simple and sufficient" The analysis shown in the supplemental (e.g. Figure S3) does not strike me as simple, especially if confronted with a peak from an unidentified compound. I do wonder how well this correction would work when a large signal on one channel is measured simultaneously with a small signal on the other. Please provide some error or uncertainty analysis for this correction.**

> While such errors are not impossible, this aspect of data quality control has been straight-forward in our experience. First, we consider the case presented by the reviewer in which one RGA measures a large signal while the second RGA measures a small signal at the exact same retention time. To mistakenly identify the real but less abundant signal as the shadow peak of the more abundant signal, the relative abundances of all major ions in the mass spectrums of both compounds would need to be similar. Since the retention times of the chromatograms of the two channels are offset by 10 min, even members of the same compound class simultaneously eluting into the two RGAs will typically have distinct mass spectra, because this retention time difference corresponds to a difference in volatility and molecular weight. In addition, both peak shape and width would need to be similar. Peak widths during retention times of 0 to 8 min of the chromatogram are generally wider (> 5 s FWHM, Fig. S10), further distinguishing them from the shadow peaks at retention times from 10 to 18 min (FWHM of 2 to 4s), and vice versa. Thus, peak width also aids in this discrimination for 80% of the chromatogram, though we find that distinct mass spectrums are sufficient.

> Even if a distinct compound was missed in a particular chromatogram due to mistaken assignment as a shadow peak, it might still be identified in a different sample for which the other RGA did not have an abundant similar signal, thus alerting the data analyst to the compound's possible existence in all samples and drawing their further scrutiny. If not, the effect of this error is that the compound would be missed and omitted from analysis, as is the case of all compounds not detected for other reasons (e.g. volatility, thermal lability, sensitivity).

> Finally, the converse case (a small shadow signal mistakenly identified as real) is unlikely to cause error as the large prime signal is readily available for reference during data analysis.

> The following has been added to the figure caption:

*"Distinct mass spectra, peak shape, and peak widths (due in part to the 10 min offset in retention time for prime and shadow chromatograms) aid in discriminating prime and shadow peaks."*

**Section 2.7 Liquid calibration standards. I didn't see any discussion of instrument zeros or blanks to challenge the system memory after measuring an ambient or calibration sample. Please provide some characterization of instrument zeros (apologies if this was somewhere in the supplemental and I just missed it).**

Thank you for suggesting the addition of this section. The new supplemental tables are included at the end of these comments:

*"3.5.1 Instrument blanks and sample carryover*
*The zero level in the calibration curve was used to evaluate contamination of instrument blanks (Table S4). Some analytes (octane, naphthalene, n,n-dimethylformamide, pentadecane) were measured in only the A1 and A2 blank samples, though this level of contamination may simply be below the detection limits for the B1 and B2 channels (Section 3.5.2). Decane was measured in blanks for all channels, and all blank measurements were less than 50 ppt except for benzene in channel A1 (123 ± 21 ppt). The high volatility of benzene suggests this might be a product of material degradation.*

*Subsequent to the highest concentration level in the calibration curve experiment, material desorbed from an additional collection from the trap was analysed to evaluate carryover (Table S5). Analyte concentration in the initial injection ranged from 2.3 to 9.2 ppb for different compounds, and subsequent carryover was less than 5% except for n-dimethylformamide (ranged from 3 ± 1 % in A2 to 26 ± 9 % in B1), 1-methylnaphthalene (13 ± 15 % in B1 only), tetradecane (20 ± 7 % in B1, 6 ± 3 % in B2), and pentadecane (ranged from 5 ± 1 % in A2 to 47 ± 11 % in B1). These compounds are all more difficult to transfer due to either volatility or functionality. The consistently superior performance of A2 and poor performance of B1 suggests that channel-specific differences such as active sites or cold spots may drive this carryover.*

**Line 260. The discussion of compounded-dependent difference in detector response is important, as the ability to intercompare measurements between channels for uncalibrated species hinges upon this. This analysis relies upon ambient data, assuming that signals are relatively constant within a 30-min window. This section would benefit if the authors could use their calibrated compounds to validate this assumption (or identify optimal time periods for this analysis), along with the utility of using the quantitation ion as the independent variable. Does the quant ion correlate with volatility? Regardless, this section requires some uncertainty analysis.**

The underlying assumption of this approach is a constant rate of change of concentration. For example, if the species concentration increases at a constant rate, then the concentration of a given sample is equal to the average of the previous and subsequent sample. The multiple iterations of the Cook's distance test are used to remove the influence of the most dynamic events which do not fulfill this assumption (e.g. the initial increase in concentration at the beginning of the stirfry), not to remove statistical outliers in a strict sense. As suggested in this comment, we have quantified the performance of this approach relative to linear calibration with liquid standards; the root mean square deviation of the compound-specific factor from the ratio of the means of the RGA B slopes and RGA A slopes was 14%.

We considered both quantification ion and boiling point as independent variables. We selected quantification ion, which unlike boiling point is known for even unknown compounds, and also correlated more strongly with the log of the compound-specific factor ($R^2 = 0.36$) than boiling point did ($R^2 = 0.20$). The correlation between quantification ion and boiling point was 0.11.

This text has been added to the end of Section 2.9:
*"The underlying assumption of this approach is a constant rate of change of concentration over three samples (30 min). For example, if the species concentration increases at a constant rate, then the concentration of a given sample is equal to the average of the previous and subsequent sample. The multiple iterations of the Cook's distance test are used to remove the influence of the most dynamic events which do not fulfill this assumption (e.g. the initial increase in concentration at the beginning of cooking), not to remove statistical outliers in a strict sense. For compounds quantified with liquid calibration standards, the root mean square deviation of the compound-*

*specific factor from the ratio of the means of the calibration curve slopes for RGA B channels and RGA A was 14%."*

**Discussion in sections 3.3 and 3.4 are well presented. One aspect of the instrument that I would have liked to have seen here is some analysis or discussion of the aging of the sample traps, specifically regarding artifact and breakthrough volumes. How often must the sample traps be replaced during ambient use? If the authors can bring in information from Wernis et al. 2021, that would be fine here.**

> In our experience, maintenance has been required for the trap itself (air leaks) or the shifting of sorbent in the trap prior to the development of significant artifacts from sorbent degradation. More systematic analysis may be possible with future work as these traps are used in this and other instruments in longer campaign. This opportunity for future work has been noted in Section 4.2: "Finally, the long-term stability and performance of the sorbent collection should be addressed in future work."

**I enjoyed the data discussion in Section 3.6, which showed a nice use of this new instrument (although see comment below regarding Figure 10). Was there any co-sampling by high-time response instrument, e.g. CIMS, CO/CO2) that could be compared with the GC data to further demonstrate the value of the continuous data measured by the GC?**

> Other high time resolution instrumentation acquiring data during the indoor measurements shown in Fig. 10 included a Dusttrak DRX ($PM_{2.5}$) and an ozone analyzer (Thermo 49i). Ozone measurements by the dual cell photometer appear to have been vulnerable to significant artifacts (possibly the high abundance of absorbing VOCs generated by the stirfry or, alternatively, shifts in relative humidity which after the reflectivity of the photometer cell). Thus, the PM2.5 timeseries, but not that of ozone, has been added to Fig. 10 (shown at the end of these responses).

**In Section 4.1, the authors discuss future improvements for the instrument, informed by the lessons learned from their first deployments. I do wonder If the stated goal of further increases sampling rate, by decreasing sample volume collected, or extending the volatility range of the GC via higher column temperature, is the best path considering the relatively high LODs for this instrument.**

> We agree with the reviewer that these potential changes have inherent disadvantages, and our goal in this section was to highlight these compromises in part to justify the current design and method. To clarify this intent, we have added the following to the start of this section: *This section considers potential challenges and compromises of modifying volatility range or increasing time resolution.*

**Section 4.2. Limitations. During this discussion of the effort required by the authors to merge the data collected on RGA A and RGA B, I had hoped that they would discuss whether the large difference in sensitivity here was inherent to the design of the paired RGAs, or just a happenstance of their detector. Did they discuss this issue with the detector manufacturer? Have they considered other means to cross-calibrate (e.g. adding a valve to systematically switch the detector that the column effluent was directed towards)?**

> Both RGAs met specification, and the difference in sensitivity was not due to the electronics units, which are easily swapped. The sensitivity difference was also observed with the separate Faraday detector, suggesting that the source and/or quadrupole were the cause of the difference. We did not attempt to swap the sources between the two instruments. We understand from the manufacturer that variability in the precision of the hardware results in varying sensitivity. Limitations of time precluded further collaborative investigation of root cause.
>
> Given the high level of complexity already inherent in MOIRA's multichannel design, we chose not to add an additional valve. Instead, measurements of chamber background as well as tuning compound were used to compare the two RGAs.

**Specific comments**

**Line 89. Specifying the size of the helium cylinder isn't needed. You could state the regulator delivery pressure if that is relevant.**

> This information has been removed.

**Line 92. "helical vibration isolators" I believe the standard name for these is "wire-rope vibration isolators".**
> This wording has been updated.

**Line 125. Can you describe the GC sample pump used?**
> The following has been added: *"The original sample pump used in this work (MPU2685-N86, KNF Neuberger, Inc.) intermittently failed during pilot mobile measurements, and it has since been replaced (NPK 09 DC, KNF Neuberger, Inc.)."*

**Line 126. "collector is purged". Please specify the direction of purge (forward- or reverse-purged). Also, can you provide the purge flow rate rather than the flush gas pressure?**
> We have specified that the *"collector is forward purged with ultra-high purity helium (28 Pa, approximately 30-40 mL min$^{-1}$)"*.

**Line 128. "maintained at 30 °C" Please provide a range (1σ) for this temperature.**
> From the indoor measurements, typical standard deviation for a 10 min collection is 0.4°C ) (1 Hz). This uncertainty has been added to the manuscript text.

**Line 151. "by transfer lines" Please specify the material of these lines, and if they are temperature-regulated before entering the vacuum chamber.**
> This information has been added: *"The GC effluent is carried into the vacuum chamber by coated stainless steel transfer lines (127 μm ID; Inertium® coating, AMCX, Inc.), which are maintained at 200°C and are parallel to the quadrupole rods, terminating just outside of the crossbeam ionization source of each RGA (Fig. S1)."*

**Line 155. "total scan time of 553 ms" Does this provide a sufficient number of data points to describe the chromatographic peaks with FWHM < 3 sec?**
> When the full width at half maximum of a Gaussian peak is 2.5 s, its height is 5% of its maximum value at a width of 4.5 s. Thus, with a scan rate of 553 ms, there are eight points with signal greater than 5% of the peak maximum. While traditional analysis recommends at least 10 points per peak, the 4 and 6% single-channel precision of the A1 and A2 measurements (even with the variability introduced by manual injection) indicate that this scan rate is acceptable. Use of the TERN in Igor software package, which fits a Gaussian curve to the single ion chromatogram peak (as opposed to numerical integration) may contribute to this measurement precision (Isaacman-Vanwertz, 2017).

**Line 160. "ionization sources" Please provide a brief description of this – filament-type? Emission current?**
> The following has been added to this section: *"The tungsten filament is operated at an emission current of 1000 μA."*

**Line 164. "A plate separates" What kind of plate? Does it have any voltage potential to reduce crosstalk?**
> Additional details have have been added to this section, and a reference to the CAD rendering of the chamber has been added: *An aluminum plate is bolted to the vacuum chamber in between the ionization sources of the two RGAs (Fig. S1).*

**Line 164. "above a threshold" Can you put some number to this threshold, preferably in ambient mixing ratio.**
> The following has been added to this section: *"As an example, in the indoor pilot measurements of limonene, RGA A and B detected shadow peaks when the prime channels measured concentrations above 20 and 59 ppb respectively."*

**Line 226. "A seven-point calibration curve was measured with a replicate at all concentration levels" While commendable, this strikes me as particularly labor-intensive and impractical for an extended deployment (e.g. continuous operation). Can you describe the time required for this (roughly 2.5 hours, I think)?**
> Yes, this was quite labor intensive (28 hours total = 7 levels * 6 replicates * 4 channels * 10 min * (1 hr / 60 min)) and intended specifically for the evaluation of instrument capabilities in this manuscript. This would not be suitable for future uses of the instrument in lab or field campaign.

**Line 234. "the daily correction factor." How much variance exists with this correction factor? Can you provide some range for this, or a time series?**

> The range has been added: "All concentration levels for a given day could then be corrected for drift by normalizing by the daily correction factor, *which ranged from 0.90 to 1.14.*"

**Line 248. "both linear and power fits were investigated here." Please provide some rationale (i.e. physical explanation) for why the instrument would have a power function sensitivity curve. Consider using only a linear calibration curve without this explanation.**

> Power fits have been used for similar thermal desorption GC-MS instruments such as the thermal desorption aerosol gas chromatograph (TAG) (Kreisberg et al., 2009). A possible physical basis is a change in transfer efficiency through the instrument with compound loading, especially near the detection limit. We have added the text in italics:
> "Applying calibration curves is one approach to addressing differences in sensitivity between the channels, and both linear and power fits were investigated here *(Kreisberg et al., 2009). Power law fits can be appropriate when the transfer of an analyte is affected by its loading such as near the detection limit.*"

**Line 274. "The inlet tubing (1/8" diameter, 2.2 m length, PFA)". Is that 1/8" diameter refer to ID or OD?**

> "*OD*" has been added to the text, and the units have been updated to metric (mm).

**Line 309. "Carboxylic acids: Acetic, propanoic, and benzoic acids were all detected during the indoor field study." Has any work been done with this compound class to determine if they are measured quantitatively? Carboxylic acids are notoriously difficult to pass through GC systems measuring ambient air.**

> We have not yet investigated the quantitative transfer of carboxylic acids through the MOIRA system. Individually heating each transfer line downstream of the sample inlet valve likely aids in sample transfer within the instrument as is the case for even larger semivolatile carboxylic acids in the similarly constructed thermal desorption aerosol gas chromatograph (TAG). However, in the case of the TAG, the acids are transported in the particle phase into the instrument collector, in contrast to gas-phase acids which may partition to sample tubing prior to the instrument inlet. We have added the following text to this section to reflect this: *"While carryover of less volatile analytes between samples is evaluated in Section 3.5.1, this analysis does not include all compound classes and notably omits carboxylic acids."*

**Line 328. "less than 3 s for most compounds with retention times greater than 900." With the detector operating at nominal 2 Hz (line 156), are enough data points collected per peak to sufficiently define the peak shape?**

> Please see the response above for the reviewer's comment regarding line 155.

**Line 334. "3.2 Precision of GC temperature control and retention time." While technically impressive, this discussion strikes me as something better suited for the supplemental. I'd suggest moving it there.**

> The precision of GC temperature control as well as its effect on retention time are crucial for successful integration of these four channels. We believe that this section is important to describing the instrument and would prefer to retain it in the main manuscript.

**Line 432. "the limit of quantification (LOQ) was calculated" I'm curious why the authors used the standard deviation of the fitted line intercept here, but used standard deviation of the baseline for LOD. Could they provide a brief rationale (or reference) for this in the text?**

> Thank you for this question, which has helped us reassess our approach to this section and figure, which were the subject of several comments by both reviewers. Our intent is to compare how the minimum concentrations detected with the instrument in the calibration curve experiment (Fig. 8c) compare with (1) the calculated limits of detection from the uncertainty of the calibration curve (Fig. 8b) and (2) the baseline noise of the MS detector (Fig. 8a). To better align these metrics for comparison, we have revised the manuscript to calculate limit of detection (Eq. 2) with both of these measures of uncertainty, eliminating reference to limit of quantification:.
>> *"Limits of detection (LOD) were calculated from two measures of uncertainty: the baseline noise $\sigma_{baseline}$ of a blank chromatogram according to Eq. (2) and the standard error $\sigma_{intercept}$ of the intercept of the calibration curve (Eq. (3)) (Fig. 8a):*
>> $$LOD_{baseline} = 3\sigma_{baseline}/S \qquad (2)$$

$$LOD_{intercept}= 3\sigma_{intercept}/S \qquad (3)"$$

The values in the text have also been updated, and no other changes to interpretation were required with this revision.

**Line 439. Figure 8. I found this figure very difficult to pull information from, especially with the small y-axis log-scaled. Could the authors consider converting this figure into a table for an easier understanding of the underlying data?**

We made several changes to this figure to increase readability including scaling each subpanel's y-axis individually (with additional gridlines) and adding space between the markers for different series. With the four channels, two dwell times, and three metrics for detection limit, a table would be quite extensive, and we hope that this revised figure has improved clarity for the reader.

**Line 457. "though a power fit is more representative at lower concentrations for some compounds" Again, please provide some physical basis for the power fit.**

We have added the following text in italics: "While the interchannel repeatability of the power law calibration curve was worse than that of the linear calibration curve, *extrapolation of* the power law *provided* better fit *of* the SIM data of some compounds at lower concentrations, which are not included in the regression. *Power law calibrations can be appropriate if the transfer of an analyte depends on its concentration, but in this case, the better fit is likely due to the inclusion of the origin in the power law (Kreisberg et al., 2009)."*

**Line 478. Figure 9. I found the image quality of Figure 9 to be poor, likely of lower resolution compared to other figures in the manuscript. Can the authors revise with a higher resolution version?**

We prepared a vector graphics version of this figure and made a few additional changes (larger legend, larger axis labels) as per the other reviewer's comments. The revised figure is included at the end of these responses.

**Figure 522. Figure 10. This is a very nice time series, but I was frustrated that no species were shown in units of mixing ratio. Can the authors provide a dual axes or stacked axes figure, presenting as many species as possible with units of mixing ratio rather than relative abundance? I feel using only relative abundance here diminishes the value of the work done here.**

This figure (shown at the end of these comments) has been updated with additional subpanels for the mixing ratios of the compounds shown which were in the calibration standard as well as particulate matter.

**Line 542. Figure 11. Can markers be added to color trace to indicate start and stop location of each sample? The discontinuous data shown here is counter to the operational goal of the instrument (relatively high-time series continuous measurements). If this was due to instrumental issues (e.g. sample pump failure), consider making some brief note to that effect in the figure caption, which may be as far as some readers will go to parse what is shown here.**

We have added the following text to the figure caption as recommended by the reviewer: *"These samples represent less than four hours of sampling time, but data recovery during this first deployment was limited by issues with the sample pump and ethernet switch which have since been resolved."* We considered showing a smaller map area with additional markers as suggested in this comment, and we preferred showing all data as in the original figure. Development of more effective visualizations of this spatially-resolved data is part of future work with more extensive mobile campaigns.

References

Isaacman-Vanwertz, G., Sueper, D. T., Aikin, K. C., Lerner, B. M., Gilman, J. B., De Gouw, J. A., Worsnop, D. R., and Goldstein, A. H.: Automated single-ion peak fitting as an efficient approach for analyzing complex chromatographic data, J. Chromatogr. A, 1529, 81–92, https://doi.org/10.1016/j.chroma.2017.11.005, 2017.

Kreisberg, N. M., Hering, S. V., Williams, B. J., Worton, D. R., and Goldstein, A. H.: Quantification of Hourly Speciated Organic Compounds in Atmospheric Aerosols, Measured by an In-Situ Thermal Desorption Aerosol Gas Chromatograph (TAG), Aerosol Sci. Technol., 43, 38–52, https://doi.org/10.1080/02786820802459583, 2009.

[Figure]

**Revised Figure 8. (a) Limit of detection (LOD$_{baseline}$) calculated from the baseline of a blank chromatogram (Eq. (2)) (b) Limit of detection (LOD$_{intercept}$) calculated from the standard error of the calibration curve slope (Eq. (3)) (c) Minimum concentration detected among calibration samples as compared to the minimum injected concentration level in the calibration experiment. Due to significant evaporation from the calibration standard, the most volatile analytes (e.g. isoprene, pentane, hexane) are not included here. Alkanes are listed by their carbon number.**

[Figure]

**Figure 9. Linear and power calibration curves (error bars ± 1 standard deviation) for (a) toluene, (b) octane, (c) 2-hexanone, and (d) 1-methylnaphthalene. For similar scale across channels, peak areas are corrected by each channel's sensitivity factor (FA1 = 1.68, FA2 = 1.83, FB1 = 0.227, FB2 =0.250), which is the median ratio (across all compounds) of the calibration curve slope for a given compound and channel and the mean slope for that compound for all four channels. The inset plot shows the lower range of data. Open circles indicate concentration levels for which no peak was detected on among at least one of the 2 ms scan replicates (n=6). Pearson R2 values include only replicates for which n=6 (whose mean is represented by the filled circles).**

[Figure]

**Revised Figure 10.** **Time series of indoor measurements of (a) relative abundance of VOCs with compound-specific correction and normalization to the maximum measurement (b) mixing ratios for compounds included in the calibration standard and (c) PM2.5. The vertical lines correspond to the following activities (with times rounded down to the nearest sampling time): 1 - Air freshener plugged in, 2 - Chopped vegetables, 3 - Heated vegetable oil, 4 - Stir-fried onion, garlic, and ginger, 5 - Stir-fried mushroom with black pepper, 6 - Washed dishes, 7 - Cleaned floors with pine cleaner and opened kitchen door, 8 - Closed kitchen door, 9 - Opened kitchen door, 10 - Closed kitchen door, 11 - Opened windows, 12 - Closed windows, heated oil and black pepper, 13 - Stir-fried onion, 14 - Stir-fried mushrooms, 15 - Opened windows, turned on vent fan, 16 - Closed windows, turned off vent fan, 17 - Cleaned floors with pine cleaner, opened kitchen door.**

[Figure]

© OpenStreetMap contributors 2023. Distributed under the Open Data Commons Open Database License (ODbL) v1.0.

**Figure 11. Map of normalized relative abundances of toluene and methyl ethyl ketone measured during pilot mobile measurements in St. Louis, Missouri (July 20-21, 2022). Gasoline and diesel fueling stations within 50 m of sampling locations are indicated. Stationary collections are represented by open circles, while samples collected while the vehicle is moving are represented as lines. As an example, the total distance of one of the mobile samples (2.2 km) is indicated in (a). These samples represent less than four hours of sampling time, but data recovery during this first deployment was limited by issues with the sample pump and ethernet switch which have since been resolved.**

**Table S4. Concentration of blank samples in calibration curve experiment by channel (ppt, mean ± standard deviation, n = 6)**

|  | A1 | A2 | B1 | B2 |
|---|---|---|---|---|
| isoprene | 0 ± 0 | 0 ± 0 | 0 ± 0 | 0 ± 0 |
| C5 | 0 ± 0 | 0 ± 0 | 0 ± 0 | 0 ± 0 |
| methyl tert-butyl ether | 0 ± 0 | 0 ± 0 | 0 ± 0 | 0 ± 0 |
| C6 | 0 ± 0 | 0 ± 0 | 0 ± 0 | 0 ± 0 |
| benzene | 123 ± 21 | 0 ± 0 | 0 ± 0 | 0 ± 0 |
| toluene | 0 ± 0 | 0 ± 0 | 0 ± 0 | 0 ± 0 |
| 4-methyl-2-pentanone | 0 ± 0 | 0 ± 0 | 0 ± 0 | 0 ± 0 |
| C8 | 24 ± 13 | 43 ± 5 | 0 ± 0 | 0 ± 0 |
| 2-hexanone | 0 ± 0 | 19 ± 48 | 0 ± 0 | 0 ± 0 |
| ethylbenzene | 0 ± 0 | 0 ± 0 | 0 ± 0 | 0 ± 0 |
| m,p-xylene | 0 ± 0 | 17 ± 43 | 0 ± 0 | 0 ± 0 |
| o-xylene | 0 ± 0 | 0 ± 0 | 0 ± 0 | 0 ± 0 |
| styrene | 0 ± 0 | 0 ± 0 | 0 ± 0 | 0 ± 0 |
| n,n-dimethylformamide | -33 ± 36 | 36 ± 39 | 0 ± 0 | 0 ± 0 |
| alpha-pinene | 0 ± 0 | 0 ± 0 | 0 ± 0 | 0 ± 0 |
| beta-pinene | 0 ± 0 | 0 ± 0 | 0 ± 0 | 0 ± 0 |
| 1,2,4-trimethylbenzene | 0 ± 0 | 0 ± 0 | 0 ± 0 | 0 ± 0 |
| C10 | 32 ± 6 | 38 ± 5 | 1 ± 17 | 20 ± 19 |
| 1,2,3-trimethylbenzene | 0 ± 0 | 0 ± 0 | 0 ± 0 | 0 ± 0 |
| limonene | 0 ± 0 | 0 ± 0 | 0 ± 0 | 0 ± 0 |
| C12 | 0 ± 0 | 0 ± 0 | 0 ± 0 | 0 ± 0 |
| naphthalene | 41 ± 3 | 24 ± 26 | 0 ± 0 | 0 ± 0 |
| 1-methylnaphthalene | 0 ± 0 | 0 ± 0 | 0 ± 0 | 0 ± 0 |
| C14 | 0 ± 0 | 0 ± 0 | 0 ± 0 | 0 ± 0 |
| C15 | 4 ± 10 | 10 ± 25 | 0 ± 0 | 0 ± 0 |

**Table S5.  Percent carryover in subsequent blank sample by channel (%, mean ± standard deviation,  n = 6)**

| | Concentration of Initial Sample (ppb) | A1 | A2 | B1 | B2 |
|---|---|---|---|---|---|
| methyl tert-butyl ether | 5.5 | 0 ± 0 | 0 ± 0 | 0 ± 0 | 0 ± 0 |
| benzene | 6.3 | 2 ± 0 | 0 ± 0 | 0 ± 0 | 0 ± 0 |
| toluene | 5.3 | 0 ± 1 | 1 ± 1 | 0 ± 0 | 0 ± 0 |
| 4-methyl-2-pentanone | 4.9 | 1 ± 1 | 2 ± 1 | 0 ± 0 | 0 ± 0 |
| C8 | 4.3 | 1 ± 0 | 1 ± 0 | 0 ± 0 | 0 ± 0 |
| 2-hexanone | 4.9 | 2 ± 0 | 3 ± 1 | 1 ± 2 | 0 ± 0 |
| ethylbenzene | 4.6 | 1 ± 1 | 2 ± 1 | 0 ± 0 | 0 ± 0 |
| m,p-xylene | 9.2 | 1 ± 0 | 1 ± 1 | 0 ± 0 | 0 ± 0 |
| o-xylene | 9.2 | 1 ± 0 | 2 ± 1 | 0 ± 0 | 0 ± 0 |
| styrene | 4.7 | 0 ± 0 | 0 ± 0 | 0 ± 0 | 0 ± 0 |
| n,n-dimethylformamide | 6.7 | 10 ± 4 | 3 ± 1 | 26 ± 9 | 12 ± 6 |
| alpha-pinene | 3.6 | 0 ± 1 | 0 ± 1 | 0 ± 0 | 0 ± 0 |
| beta-pinene | 3.6 | 0 ± 0 | 0 ± 1 | 0 ± 0 | 0 ± 0 |
| 1,2,4-trimethylbenzene | 4.1 | 1 ± 1 | 2 ± 1 | 0 ± 0 | 0 ± 0 |
| C10 | 3.4 | 2 ± 0 | 2 ± 0 | 1 ± 1 | 1 ± 1 |
| 1,2,3-trimethylbenzene | 4.1 | 2 ± 1 | 1 ± 1 | 0 ± 0 | 0 ± 0 |
| limonene | 3.6 | 1 ± 1 | 1 ± 1 | 0 ± 0 | 0 ± 0 |
| C12 | 2.9 | 2 ± 0 | 2 ± 1 | 3 ± 2 | 0 ± 0 |
| naphthalene | 3.8 | 2 ± 0 | 2 ± 0 | 1 ± 1 | 0 ± 0 |
| 1-methylnaphthalene | 3.4 | 3 ± 1 | 2 ± 0 | 13 ± 15 | 0 ± 0 |
| C14 | 2.5 | 4 ± 2 | 3 ± 0 | 20 ± 7 | 6 ± 3 |
| C15 | 2.3 | 11 ± 6 | 5 ± 1 | 47 ± 11 | 10 ± 6 |